# Microbial Community Abundance and Metabolism Close to the Ice-Water Interface of the Blomstrandbreen Glacier (Kongsfjorden, Svalbard): A Sampling Survey Using an Unmanned Autonomous Vehicle

Maria Papale [1,*], Gabriella Caruso [1], Giovanna Maimone [1], Rosabruna La Ferla [1], Angelina Lo Giudice [1], Alessandro Ciro Rappazzo [1], Alessandro Cosenza [1], Filippo Azzaro [1], Roberta Ferretti [2], Rodolfo Paranhos [3], Anderson Souza Cabral [3,4], Massimo Caccia [2], Angelo Odetti [2], Giuseppe Zappalà [5], Gabriele Bruzzone [2] and Maurizio Azzaro [1]

1 Institute of Polar Sciences, National Research Council (CNR-ISP), Spianata S. Raineri 86, 98122 Messina, Italy
2 Institute of Marine Engineering, National Research Council (CNR-INM), Via De Marini 6, 16149 Genoa, Italy
3 Laboratorio de Hidrobiologia, Institute of Biology, Federal University of Rio de Janeiro (UFRJ), Ilha do Fundao, Rio de Janeiro 21941-590, Brazil
4 Institute of Microbiology, Federal University of Rio de Janeiro (UFRJ), Av. Carlos Chagas Filho, Rio de Janeiro 21941-902, Brazil
5 Institute of Biological Resources and Marine Biotechnologies, National Research Council (CNR-IRBIM), 98122 Messina, Italy
* Correspondence: maria.papale@cnr.it

**Abstract:** Polar marine environments host a complex assemblage of cold-adapted auto- and heterotrophic microorganisms that affect water biogeochemistry and ecosystem functions. However, due to logistical difficulties, remote regions like those in close proximity to glaciers have received little attention, resulting in a paucity of microbiological data. To fill these gaps and obtain novel insights into microbial structure and function in Arctic regions, a survey of microbial communities in an area close to the Blomstrandbreen glacier in Kongsfjorden (Svalbard Archipelago; Arctic Ocean) was carried out during an early summer period. An Unmanned Autonomous Vehicle designed to safely obtain seawater samples from offshore-glacier transects (PROTEUS, Portable RObotic Technology for Unmanned Surveys) was equipped with an automatic remotely-controlled water multi-sampler so that it could sample just beneath the glacier, where access from the sea is difficult and dangerous. The samples were analysed by image analysis for the abundance of total prokaryotes, viable and respiring cells, their morphological traits and biomass; by flow cytometry for autotrophic and prokaryotic cells (with high and low nucleic acid contents) as well as virus-like particle counts; by BIOLOG ECOPLATES for potential community metabolism; and by fluorimetry for potential enzymatic activity rates on organic polymers. Contextually, the main physical and chemical (temperature, salinity, pH, dissolved oxygen and nutrients) parameters were detected. Altogether, besides the PROTEUS vehicle's suitability for collecting samples from otherwise inaccessible sites, the multivariate analysis of the overall dataset allowed the identification of three main sub-regions differently affected by the haline gradient (close to the glacier) or terrigenous inputs coming from the coast. A complex microbiological scenario was depicted by different patterns of microbial abundance and metabolism among the transects, suggesting that ice melting and Atlantic water inflow differently supported microbial growth.

**Keywords:** microbial abundance; microbial activity; unmanned autonomous vehicle; glacier; Svalbard

## 1. Introduction

Polar marine environments host a complex assemblage of cold-adapted auto- and heterotrophic microorganisms, representing the largest and most functionally active community and the first link in the trophic chains of these harsh aquatic systems [1]. Marine

microbes are key organisms in the biological carbon pump processes and assume a significant role in a changing global context [2,3]. In the Arctic Ocean, microbial communities are shaped by the occurrence of complex hydrological features (i.e., ice-free and ice-covered surface waters, temperature approaching freezing point values at depth layers, extreme seasonality and strong light variations) that determine fluctuations in the dominant bacterial and archaeal taxonomic groups [4,5], as well as in microbial abundance, diversity and metabolism [6]. The ongoing climate warming in the Arctic potentially affects large-scale microbial processes. In particular, glaciers retreat increases meltwater outflow and input of suspended sediment load to the coastal marine environment [7]. Currently, the increasing influence of Atlantic water into the Arctic Ocean makes it warmer and saltier, affecting the microbial community structure [8]. The Kongsfjorden is an Arctic glacial fjord located on the west coast of the Svalbard Archipelago at 79° N, 12′ E. The glaciers that reach the fjord appear as high ice walls placed almost vertically on the sea. Clearly, their conformation and continuous dynamism make the ice/sea interface almost totally inaccessible to conventional sampling methods [6,9]. To date, there is very little information on the chemical, biological and ecological dynamism of these interfaces due to the difficulties in working near the ice fronts, where taking in situ measurements is extremely hazardous. In this context, how the various levels of such marine trophic networks modify their structure and function is not yet known [6,10,11]. The development of an automatic vehicle equipped with a water multi-sampler provided us with an excellent tool for tracking the short-term variability of environmental and biological parameters at these generally inaccessible sites [12,13]. Technological advances further provide suitable tools for exploring the microbiomes under contrasting seasonal conditions, as it is documented by a previous study using an autonomous sampling approach [14].

Recently, PROTEUS (Portable RObotic TEchnology for Unmanned Surveys), a marine robotic vehicle, was developed to collect atmospheric and sea-ice data in areas that are not safely accessible to humans, such as along glacier fronts [15,16]. The use of the PROTEUS allowed us to collect water samples from stations located in the Kongsfjorden close to the Blomstrandbreen glacier front, along four different transects.

The rationale of our study is that the ocean water inflows and the glacial melt waters may have strong ecological implications for the microbial community. To address this, during the early summer (June) of 2017, a survey was carried out to evaluate the planktonic abundance (picophyto-, bacterio-, and virioplankton) and the microbial community composition and metabolism. Using an interdisciplinary approach, linking microbial and chemical-physical observations, we aimed to fill current knowledge gaps in the response of microbial abundance, structure and function to the complexity of phenomena (ice-melting and ocean water inflow) in the proximity of glaciers.

## 2. Materials and Methods

### 2.1. Study Area and Seawater Sampling by PROTEUS

The Blomstrandbreen glacier is located in the northeastern area of the Kongsfjorden, an area affected both by freshwater inputs coming from the melting glacier and by Atlantic currents from the north [17]. Recently, water circulation within the Kongsfjorden has changed following the opening of a channel between the land, the glacier, and the Blomstrandhalvøya island.

During a survey from 21 to 23 June 2017, surface water samples were collected at sampling stations located in Kongsfjorden close to the Blomstrandbreen glacier. Four transects (namely 1, 2, 3 and 4) from the glacier to the open sea were tracked (Figure 1 and Table 1). Transect 1 was sampled on 21 June and the others on 23 June.

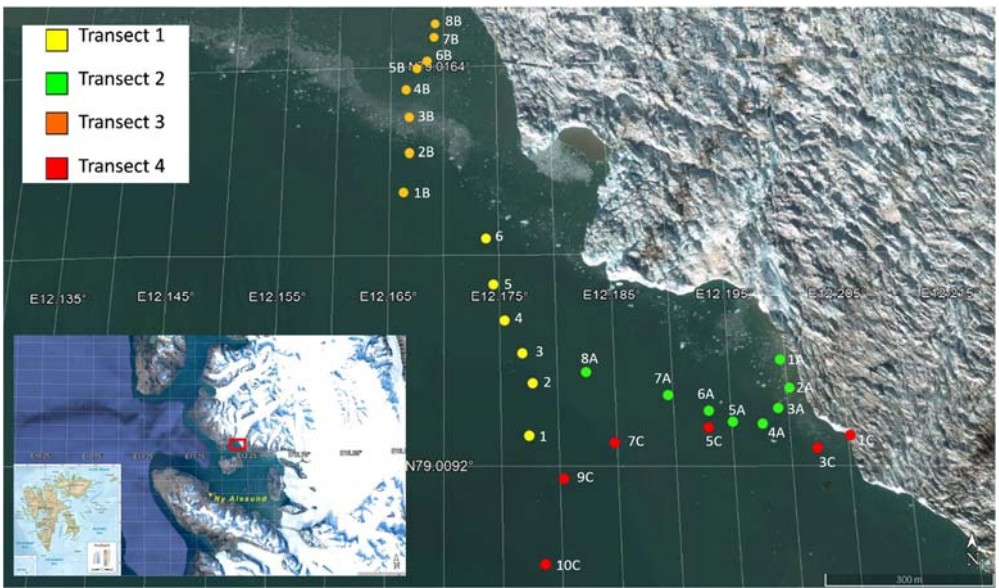

**Figure 1.** Location of the sampling stations across four glacier-offshore transects.

**Table 1.** List of the sampled stations with indications of their distances from the glacier.

| Transect 1 | | Transect 2 | | Transect 3 | | Transect 4 | |
|---|---|---|---|---|---|---|---|
| **Stations** | **Glacier Distance (m)** | **Stations** | **Glacier Distance (m)** | **Stations** | **Glacier Distance (m)** | **Stations** | **Glacier Distance (m)** |
| 6 | 286.2 | 1A | 6.7 | 8B | 216.6 | 1C | 0.8 |
| 5 | 322.54 | 2A | 10.91 | 7B | 247.47 | 3C | 49.13 |
| 4 | 381.74 | 3A | 45.16 | 6B | 309.89 | 5C | 177.52 |
| 3 | 418.71 | 4A | 82.4 | 5B | 378.52 | 7C | 351.89 |
| 2 | 423.69 | 5A | 133.21 | 4B | 405.7 | 9C | 462.47 |
| 1 | 483.30 | 6A | 167.24 | 3B | 361.69 | 10C | 562.85 |
| | | 7A | 222.13 | 2B | 388.48 | | |
| | | 8A | 321.8 | 1B | 403.38 | | |

## 2.2. The PROTEUS Unmanned Autonomous Vehicle

The UAV PROTEUS (Figure 2) is a portable, highly modular and reconfigurable vehicle. It is an innovative, highly modifiable core vehicle with various portable modules that can be added and assembled in the field to handle different tasks. PROTEUS is mainly used as a UAV and can be remotely controlled using a guidance camera mounted on top of the vehicle. At the time of sampling, PROTEUS was equipped with a water multi-sampler composed of six sterile 250 mL bottles that were controlled by a dedicated hardware and software architecture, able to carry out sampling sequentially along a transect. A data logger allowed us to follow PROTEUS's path and operational behaviour by tracing the GPS position and data from the navigation sensors, thus allowing georeferencing of the sampling phase. All the detailed features have been previously reported [15,16].

## 2.3. Physical and Chemical Parameters

An Ocean Seven 305 CTD multiprobe sensor (Idronaut, Brugherio, Italy) was used to measure temperature (T), salinity (S), pH, and dissolved oxygen ($O_2$) values.

Water samples for nutrient determinations [(ammonia ($NH_4^+$), nitrite + nitrate ($NO_2^- + NO_3^-$) and orthophosphate ($PO_4^{3-}$)] were filtered using GF/F glass filters (Whatman, Maidstone, UK) and kept frozen ($-20\ ^\circ$C). Analytical determinations were performed by a Varian Mod Cary 50 spectrophotometer, according to Strickland and Parsons [18], except for $NH_4^+$ [19].

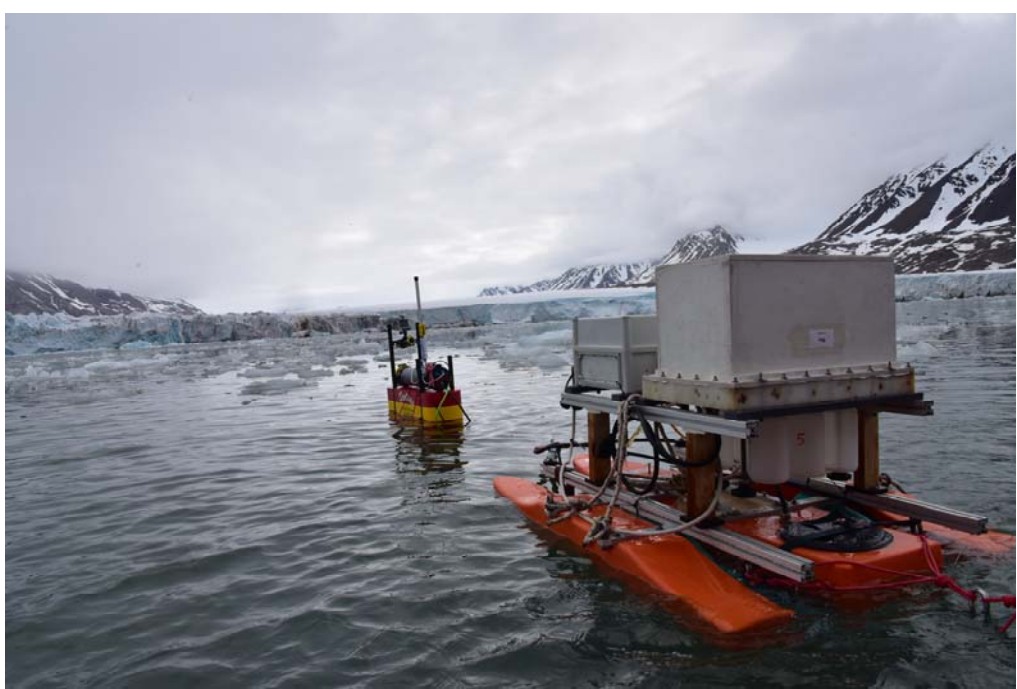

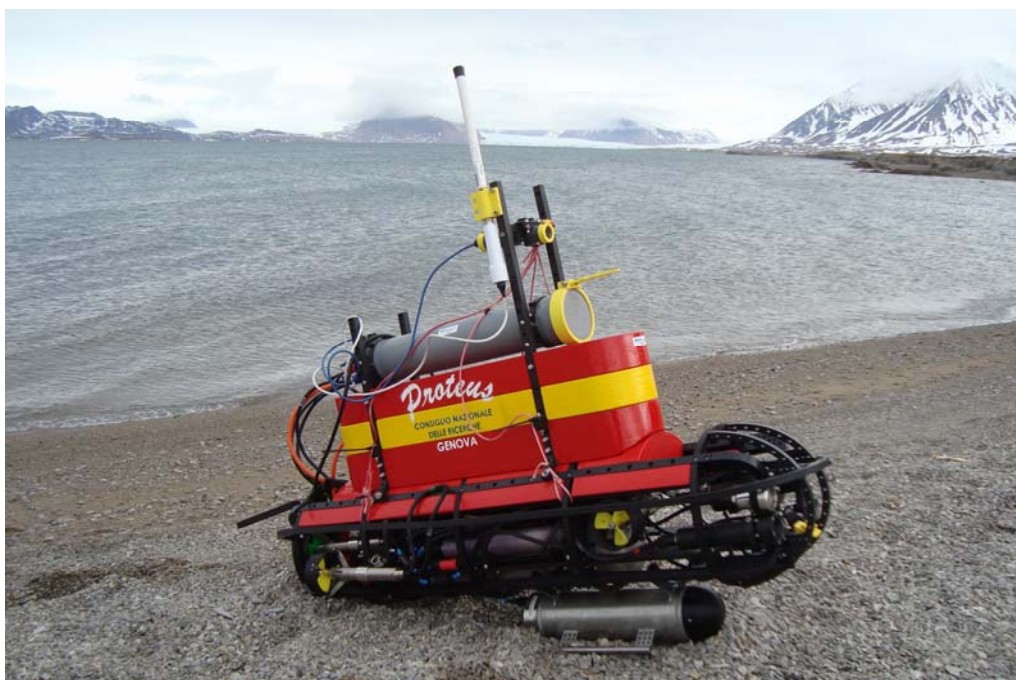

**Figure 2.** In the figure below are the UAV PROTEUS and the multi-sampler.

### 2.4. Microbial Abundances

2.4.1. Prokaryotic Cell Abundance, Volume and Biomass by Image Analysis

Samples for prokaryotic cell abundance (PA), biomass (PB) and volume (VOL) were fixed with formaldehyde (2% final concentration) and stored in the dark at 4 °C until analysis. Three replicates were filtered through black polycarbonate membranes (pore size 0.22 μm; GE Water & Process Technologies, Feasterville Trevose, PA, USA) and stained for 10–20 min with DAPI (4′6-diamidino-2-phenylindole from Sigma-Aldrich, St. Louis, MO, USA) at a final concentration of 10 μg mL$^{-1}$ [20]. The prokaryotic cells were quantified by an AXIOPLAN 2 Imaging microscope (Carl Zeiss, Oberkochen, Germany), provided with the specific sets for DAPI (G365; FT395; LP420) and equipped with an

AXIOCAMHR digital camera (Carl Zeiss) and AXIOVISION 3.1 software. Cell counts were performed on a minimum of 20 randomly selected fields on three replicate slides. When cell abundance was low, 30–40 random microscope fields were counted. The detailed methodological procedures for the single-cell volume calculation and the volume-to-biomass conversion factor were reported by La Ferla et al. [21].

### 2.4.2. Quantification of Viable (Live/Dead) and Respiring (CTC+) Prokaryotic Cells

The viability of prokaryotic cells was quantified by the Live/Dead Bac Light Bacterial Viability Kit™ (Molecular Probes, Eugene, OR, USA) (L/D) as previously described by La Ferla et al. [22]. The cells were counted using the same Imaging microscope (AXIOPLAN 2, Carl Zeiss), equipped with the specific filter sets for fluoresceine (BP450-490; FT510; LP520) and rhodamine (BP546/12; FT580; LP590). The viability of prokaryotic cells—as the number of respiring cells with respect to the total abundance—was quantified by the Bac Light Redox Sensor CTC Vitality Kit™ (Molecular Probes). The procedure reported by La Ferla et al. [22] was followed.

### 2.4.3. Virus-like Particles, Autotrophic and Prokaryotic Cells by Flow Cytometry

Flow cytometry analysis was performed using a FACSCalibur flow cytometer (BD-Becton, Dickinson and Company, Franklin Lakes, NJ, USA) with the standard laser and optics: i.e., an air-cooled argon-ion laser emitting at 488 nm (power at 15 mW), fixed laser alignment, and fixed optical components. A 70-μm nozzle aspirated the sample, and sterile Mill-Q water (18.2 mΩ) was used as the sheath fluid. For the determination of viral abundance (virus-like particles, VLP), 1.5 mL of sample were collected in cryogenic tubes, fixed with glutaraldehyde (0.05%) for 30 min at 4 °C, frozen and stored in liquid nitrogen until analyses. Samples for prokaryotic counts ($PA^C$) were also fixed immediately after collection (paraformaldehyde 1% + glutaraldehyde 0.05%) and frozen in liquid nitrogen, where they were kept until analysis [23]. For viral and prokaryotes enumeration, aliquots were stained with SYBR Green I (Molecular Probes, at a final dilution of $5 \times 10^{-4}$ of the commercial stock solution) according to Brussaard [24] and analysed. Distinct viral groups and prokaryotic heterotrophic cells with high (HNA) and low (LNA) nucleic acid content were detected and quantified based on their cytometric signatures in a side scatter plot (X-axis, related by size) versus green fluorescence (Y-axis, green fluorescence from SYBR Green I related to nucleic acid content). Distinct photoautotrophic populations were detected, identified, and quantified using a combination of side scatter light and natural fluorescence (red and orange) produced by photosynthetic pigments [25].

### 2.5. Culturable Heterotrophic Bacteria

The fraction of culturable heterotrophic bacteria was counted by the culture method on plates of Marine agar (Conda Laboratories, Torrejon de Ardoz, Madrid, Spain), inoculated in duplicate with 0.1 mL of seawater on their surface and incubated at 4 °C for 20 days. This medium is specific for the growth of these microorganisms; their abundance was reported as colony-forming units (CFU) per mL of sample.

### 2.6. Prokaryotic Community Composition

The in-situ abundance of different phylogenetic groups was performed by the CAtalyzed Reporter Deposition—Fluorescence in Situ Hybridization (CARD-FISH) technique [26] for samples from Transects 1, 2 and 4. Samples from Transect 3, pretreated in situ for CARD FISH, were damaged during transport back to Italy, and it was not possible to analyse them. A sample volume ranging from 3 and 6 mL, depending on the DAPI hybridisation results, was filtered on white polycarbonate membranes (diameter, 25 mm; pore size, 0.22 μm) and subsequently fixed for 30 min at room temperature by overlaying filters with a freshly prepared paraformaldehyde (final concentration 4%)—phosphate-buffered saline (PBS, 130 mM NaCl, 10 mM $NaHPO_4$, and 10 mM $NaH_2PO_4$, pH 7.4) solution (3 mL). After fixative removal, the membranes were washed twice with 3 mL of PBS and, finally, with distilled water.

Filter sections were then hybridised with horseradish peroxidase (HRP)-labelled oligonucleotide probes (listed in Table 2). According to the protocol designed by Pernthaler et al. [27], the tyramide signal amplification was carried out. The probe NON338 [28] was used as a negative control, and no false positive signals were detected. Filters were counterstained with 4,6-diamidino-2-phenylindole (DAPI) at a final concentration of 1 μg mL$^{-1}$ and subsequently mounted on glass slides with Citifluor and Vectashield in a 4:1 proportion and enumerated by an Axioplan epifluorescence microscope (Carl Zeiss) equipped with specific filter sets for DAPI and HRP. For all the probes used, the actual counts were done by enumerating a minimum of 500 DAPI-stained cells in 20 fields at least to cover an area of 100 μm × 100 μm each.

**Table 2.** HRP-labelled oligonucleotide probes used in this study. Probes EUB338I, EUB338II and EUB338III were equimolarly mixed together to obtain the EUB mix; probes DELTA495a, DELTA495b and DELTA495c were equimolarly mixed together to obtain the DELTA mix.

| Probes | Target Group | Probe Sequence (5′–3′) | References |
|---|---|---|---|
| EUB338I | Most, but not all, bacteria | GCTGCCTCCCGTAGGAGT | [29] |
| EUB338II | Planctomycetes | GCAGCCACCCGTAGGTGT | [30] |
| EUB338III | Verrucomicrobiales | GCTGCCACCCGTAGGTGT | [30] |
| ALF968 | Alphaproteobacteria | GGTAAGGTTCTGCGCGTT | [31] |
| BET42a | Betaproteobacteria | GCCTTCCCACTTCGTTT | [32] |
| GAM42a | Gammaproteobacteria | GCCTTCCCACATCGTTT | [32] |
| DELTA495a | Most Deltaproteobacteria | AGTTAGCCGGTGCTTCCT | [33] |
| DELTA495b | Deltaproteobacteria | AGTTAGCCGGCGCTTCCT | [33] |
| DELTA495c | Deltaproteobacteria | AATTAGCCGGTGCTTCCT | [33] |
| EPSY914 | Epsilonproteobacteria | GGTCCCCGTCTATTCCTT | [34] |
| CF319a | Bacteroidetes | TGGTCCGTGTCTCAGTAC | [35] |
| ARCH915 | Archaea | GTGCTCCCCCGCCAATTCCT | [36] |

*2.7. Microbial Metabolism*

2.7.1. Physiological Profiles of Microbial Community

Following the standardised procedures [37,38], Biolog-Ecoplates™ (Rigel Life Sciences, Rome, Italy) were used to determine the metabolic potentials of the microbial assemblages hosted in the samples and their differences. Ninety-six-well microtiter plates containing 31 carbon sources and a control in triplicate, together with the redox dye tetrazolium violet, were inoculated with 150 μL of the sample. Each plate was incubated at 4 °C in the dark [22]. The use of carbon sources as energy, and the consequent respiratory activity of the microorganisms, allowed the formation of formazan, and the response was measured as the absorbance at 590 nm using a microplate-reader spectrophotometer (MICROTITER ELX-808, Bio Whittaker, Inc., Walkersville, MD, USA) equipped with an Automatic Microplate Reader and specific software (WIN KQCL) for data processing. The optical density (OD) was recorded immediately after inoculation (time T0) and every 2 days. According to Sala et al. [39], the colour development for each plate was expressed as the average substrate colour development (ASCD) using the following equation by Sala et al. [40]:

$$ASCD = \Sigma \left( (R - C)/31 \right)$$

where R was the average absorbance of the three wells with substrate and C was the average absorbance of the control wells (without substrate). The absorbance percentages of each substrate were determined according to Sala et al. [40], and a value of 2% of the total absorbance measured per plate was used as the threshold value for substrate utilisation. The carbon substrates were grouped into six guilds, i.e., complex carbon sources (polymers), carbohydrates, phosphate carbon sources, carboxylic and acetic acids, amino acids, and amines.

### 2.7.2. Enzymatic Activities

The three enzymes, such as leucine aminopeptidase (LAP), ß-glucosidase (ß-GLU) and alkaline phosphatase (AP)—involved in the breakdown of proteins, polysaccharides and organic phosphates, respectively—were measured as their potential hydrolytic activity rates towards organic polymers as exhibited by the microbial community. The enzymatic assays were carried out using the fluorogenic substrate addition method [41]. The natural compound analogues L-leucine-7-amido-4-methylcoumarin hydrochloride, 4-methylumbelliferyl-B-D-glucopyranoside and 4-methylumbelliferyl phosphate (Sigma-Aldrich, Merck, Milan, Italy) were added to seawater samples for LAP, ß-GLU and AP measurements, respectively. Triplicate 5-mL sub-volumes of each sample were incubated with increasing amounts (40–320 nmol $L^{-1}$) of each substrate; fluorescence measurements were recorded at T0, just after the substrate addition, and at 3 h after incubation at 4 °C with a fluorimeter (Jenway model 6280, Staffordshire, UK) at 365 nm and 440 excitation and emission wavelengths for MCA substrates, respectively and at 365 nm and 455 nm for MUF substrates, respectively. The fluorimeter was calibrated against standards of 7-amino-4-methylcoumarin (MCA) or 4-methylumbelliferone (MUF) at concentrations of 200–800 nmoles. Enzyme activity rates were reported as the maximum hydrolysis reaction velocity ($V_{max}$) of the substrates, as calculated by a Lineweawer-Burke transformation of the reciprocal of the substrate concentrations plotted versus the reaction velocity. Data were reported as nanomoles of leucine, glucoside and $PO_4^{3-}$ potentially released per litre per hour (nmol $L^{-1}$ $h^{-1}$).

### 2.8. Statistical Analyses

All data were opportunely transformed (log or square root transformation, depending on the kind of data) to perform statistical analysis. Analysis of Variance (ANOVA) was performed to detect the statistical significance of the differences observed between the transects. Spearman correlation coefficients between microbial counts and physical, chemical and environmental parameters were calculated to establish the occurrence of significant relationships; results were considered statistically significant when $p < 0.05$. Transformed biological data were processed by calculating the Bray–Curtis similarity. The similarity matrix was used to compute the non-metric multidimensional analysis (nMDS) on which vectors from diversity data were superimposed, to observe the grouping according to the examined parameters. As a second step, the Euclidean distance was calculated, clustering analysis was applied, and a Principal Component Analysis (PCA) was carried out with the obtained resemblance matrix. The occurrence of significant differences between the four sampled transects, and all analysed parameters were tested by a Linear Regression model (LM); the results were considered statistically significant when $p < 0.05$. All analyses were performed by the R program (version 4.2.0) using the packages HTSfilter, Pheatmap Vegan and Maaslin2. The Shannon index of diversity (H') was applied to the utilisation of individual carbon sources—as a percentage of the total absorbance of the microplate for the stations of each transect, using the package Primer 6 [42].

## 3. Results

### 3.1. Environmental Parameters

The physical and chemical features recorded in each transect are reported in Table 3.

In Figure 3, the behaviour of T/S diagrams across the four transects is shown. High variability was observed for T, with minimum values at Transect 1 and maximum values at Transect 4. T ranged from 3.4 to 5.7 °C (Transect 1), from 3.6 to 6.3 °C (Transect 2), from 4.6 to 6.1 °C (Transect 3) and from 5.2 to 6.4 °C (Transect 4). Within the transects, temperature variability was also observed, with minima at stations 3 and 5, 1A, 6B and 7B, 7C and 1C, with most of them corresponding to near or at the glacier front. Spatial variability was significant, as determined by ANOVA, between Transect 1 and 4 (df = 2; F = 34.8; $p = 1.66 \times 10^{-9}$).

**Table 3.** Physical and chemical characteristics recorded across the four transects.

| | Stations | T | S | O$_2$ | pH | PO$_4^{3-}$ | NH$_4^+$ | NO$_2^-$ + NO$_3^-$ |
|---|---|---|---|---|---|---|---|---|
| | | °C | psu | ppm | | µM | µM | µM |
| Transect 1 | 6 | 4.31 | 31.967 | 9.68 | 8.39 | 0.30 | 2.16 | 1.81 |
| | 5 | 3.69 | 31.64 | 9.53 | 8.41 | 0.33 | 1.51 | 1.69 |
| | 4 | 4.20 | 31.49 | 9.63 | 8.40 | 0.23 | 1.38 | 1.86 |
| | 3 | 3.42 | 30.97 | 9.33 | 8.41 | 0.27 | 1.49 | 1.79 |
| | 2 | 3.98 | 31.43 | 9.60 | 8.39 | 0.29 | 2.60 | 3.87 |
| Transect 2 | *1* | 5.67 | 32.11 | 10.04 | 8.37 | 0.43 | 1.36 | 6.62 |
| | 1A | 3.59 | 31.21 | 10.99 | 8.40 | 0.20 | 2.69 | 2.00 |
| | 2A | 6.20 | 31.08 | 11.74 | 8.37 | 0.22 | 1.23 | 3.00 |
| | 3A | 6.23 | 31.23 | 11.70 | 8.37 | 0.34 | 1.09 | 5.00 |
| | 4A | 6.27 | 30,99 | 11.79 | 8.37 | 0.24 | 0.90 | 3.38 |
| | 5A | 6.28 | 30.90 | 11.78 | 8.37 | 0.23 | 0.93 | 2.99 |
| | 6A | 6.25 | 31.11 | 11.74 | 8.37 | 0.26 | 1.70 | 3.14 |
| | 7A | 6.17 | 31.11 | 11.76 | 8.37 | 0.64 | 1.31 | 4.37 |
| | 8A | 5.93 | 31.43 | 11.63 | 8.37 | 0.41 | 1.61 | 4.62 |
| Transect 3 | 8B | 5.65 | 31.92 | 10.53 | 8.38 | 0.2 | 1.47 | 3.64 |
| | 7B | 4.61 | 31.66 | 11.09 | 8.38 | 0.32 | 1.35 | 3.28 |
| | 6B | 4.61 | 31.64 | 11.09 | 8.37 | 0.28 | 1.36 | 3.17 |
| | 5B | 6.01 | 31.63 | 11.68 | 8.37 | 0.25 | 1.78 | 3.01 |
| | 4B | 5.97 | 24.95 | 12.34 | 8.37 | 0.36 | 1.42 | 4.22 |
| | 3B | 5.48 | 25.52 | 12.18 | 8.35 | 0.22 | 1.43 | 3.32 |
| | 2B | 6.126 | 25.33 | 12.29 | 8.36 | 0.24 | 1.56 | 2.81 |
| | 1B | 5.94 | 25.81 | 12.21 | 8.35 | 0.26 | 1.13 | 4.64 |
| Transect 4 | 1C | 5.17 | 31.78 | 11.19 | 8.39 | 0.17 | 1.44 | 6.7 |
| | 3C | 6.32 | 31.80 | 11.53 | 8.37 | 0.39 | 2.61 | 4.07 |
| | 5C | 6.41 | 31.67 | 11.53 | 8.36 | 0.22 | 1.21 | 2.04 |
| | 7C | 5.70 | 31.98 | 11.48 | 8.38 | 0.32 | 1.81 | 4.17 |
| | 9C | 6.39 | 31.81 | 11.66 | 8.36 | 0.27 | 2.145 | 4.25 |
| | 10C | 6.39 | 33.12 | 11.59 | 8.36 | 0.21 | 1.625 | 3.14 |

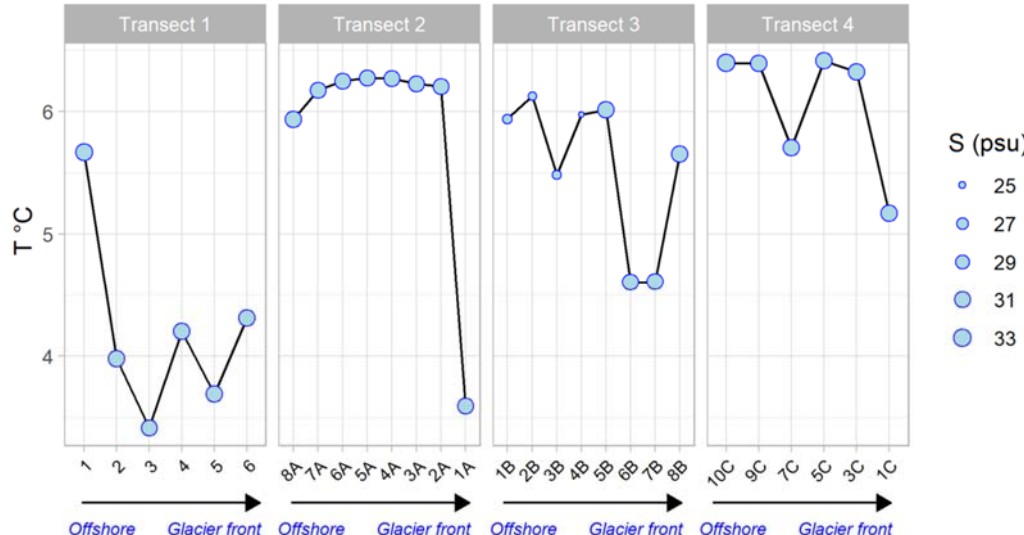

**Figure 3.** Trends of temperature and salinity values recorded across the four transects. In the *x*-axis, stations are ordered from offshore to glacier direction. The lines indicate the temperature variations, while the size of the dots indicates the variability recorded in the salinity values.

Generally, S showed homogeneous values (around 31 psu), except for transect 3 from stations 1B to 4B, where low values (around 25–27 psu) were recorded. Spatial variability was significant, as highlighted by ANOVA, between Transect 3 and 4 (df = 2; F = 37.4;

$p = 5.32 \times 10^{-5}$). Overall, all dissolved nutrients (except for $NH_4^+$) were low, showing similar trends in the four transects. The average concentration of $PO_4^{3-}$ measured in the study area was $0.29 \pm 0.1$ μM with the highest value at station 7A (0.64 μM) and the lowest value at station 1C (0.17 μM). For $NH_4^+$, the overall average concentration was $1.58 \pm 0.47$ μM, with maximum and minimum values at stations 1A (2.69 μM) and station 4A (0.90 μM). The average concentration of $NO_2^- + NO_3^-$ was $3.52 \pm 1.30$ μM. The highest value was observed at station 1C (6.70 μM) and the lowest at station 5 (1.69 μM) (Table 2). In Transects 1, 2 and 3, an increase in $NO_2^- + NO_3^-$ was from the glacier stations (6 value 1.81; 1A value 2.00; 8B value 3.64) to the stations furthest from the glacier (1 value 6.63; 8A value 4.63; 1B value 4.64), whereas in Transect 4, there was an inverted trend, the station closest to the glacier had the highest value (1C value 6.70), and the farthest showed the lowest value (10C value 3.14). ANOVA didn't show significant results regarding $NH_4^+$, while the concentrations of $NO_2^- + NO_3^-$ were significantly different between Transects 1 and 4 (df = 2; F = 67.6; $p = 2.51 \times 10^{-7}$).

*3.2. Microbial Cell Abundances*

3.2.1. Prokaryotic Cell Abundance, Volume and Biomass by Image Analysis

In Figure 4, the distribution of PA, VOL and PB for each transect and station is reported. PA was in the order of $10^6$ cells $mL^{-1}$ and ranged between 0.37 and 3.96 cells $\times 10^6$ $mL^{-1}$. The highest values were detected in Transect 1 (mean value $3.2 \pm 0.5$ cells $\times 10^6$ $mL^{-1}$) and the lowest in Transect 3 (mean value $0.6 \pm 0.2$ cells $\times 10^6$ $mL^{-1}$). Quantitative differences between the examined stations were observed, particularly in Transects 2 and 4, with a coefficient of variability (CV) of 66 and 55%, respectively. In the entire data set, VOL (about 3500 measured cells) ranged from 0.053 to 0.101 μm$^3$ (mean value $0.077 \pm 0.01$ μm$^3$) and depicted a homogeneous distribution in each transect. Contrarily to PA, the cells showed the largest VOL in Transects 2 and 3 with a similar range ($0.083 \pm 0.008$ and $0.081 \pm 0.012$ μm$^3$, respectively), intermediate values in Transect 4 ($0.076 \pm 0.008$ μm$^3$), and the lowest in Transect 1 ($0.059 \pm 0.008$ μm$^3$). PB ranged between 8.5 and 97.1 μg C $L^{-1}$ with a mean value of $41.8 \pm 26.6$ μg C $L^{-1}$. PB seemed to be more dependent on PA than VOL, following the prokaryotic abundance distribution with the highest values in Transect 1 (mean value $60.8 \pm 13.2$ μg C $L^{-1}$) and the lowest in Transect 3 (mean value $15.1 \pm 4.8$ μg C $L^{-1}$). Also, in this case, differences in terms of biomass among the stations were observed in Transects 2 and 4 (CV = 60.8 and 57.2%, respectively).

3.2.2. Evaluation of Viable (Live/Dead) and Respiring Prokaryotic Cells (CTC+)

In Table 4, the viable and respiring cells are reported.

The highest percentage of viable cells by L/D was observed in Transect 4, whilst the lowest was in Transect 2. Overall, the percentages of live cells exhibited a strong variability among the investigated stations. In Transect 1, the CV (coefficient of variability) was 155.3%. In Transects 2 and 3, the CVs were 53.1 and 111.2%, respectively. Finally, Transect 4 showed a CV of 125.1%.

Respiring cells (CTC+) were in the order of $10^4$ cells $mL^{-1}$, with a greater mean percentage in Transects 2 and 4 ($3.6 \pm 0.7$ and $3.1 \pm 1.9$% of total counts obtained by DAPI staining, respectively). Moreover, among all the stations examined, the highest variability was observed in Transects 1 and 4 (CV = 49 and 62%).

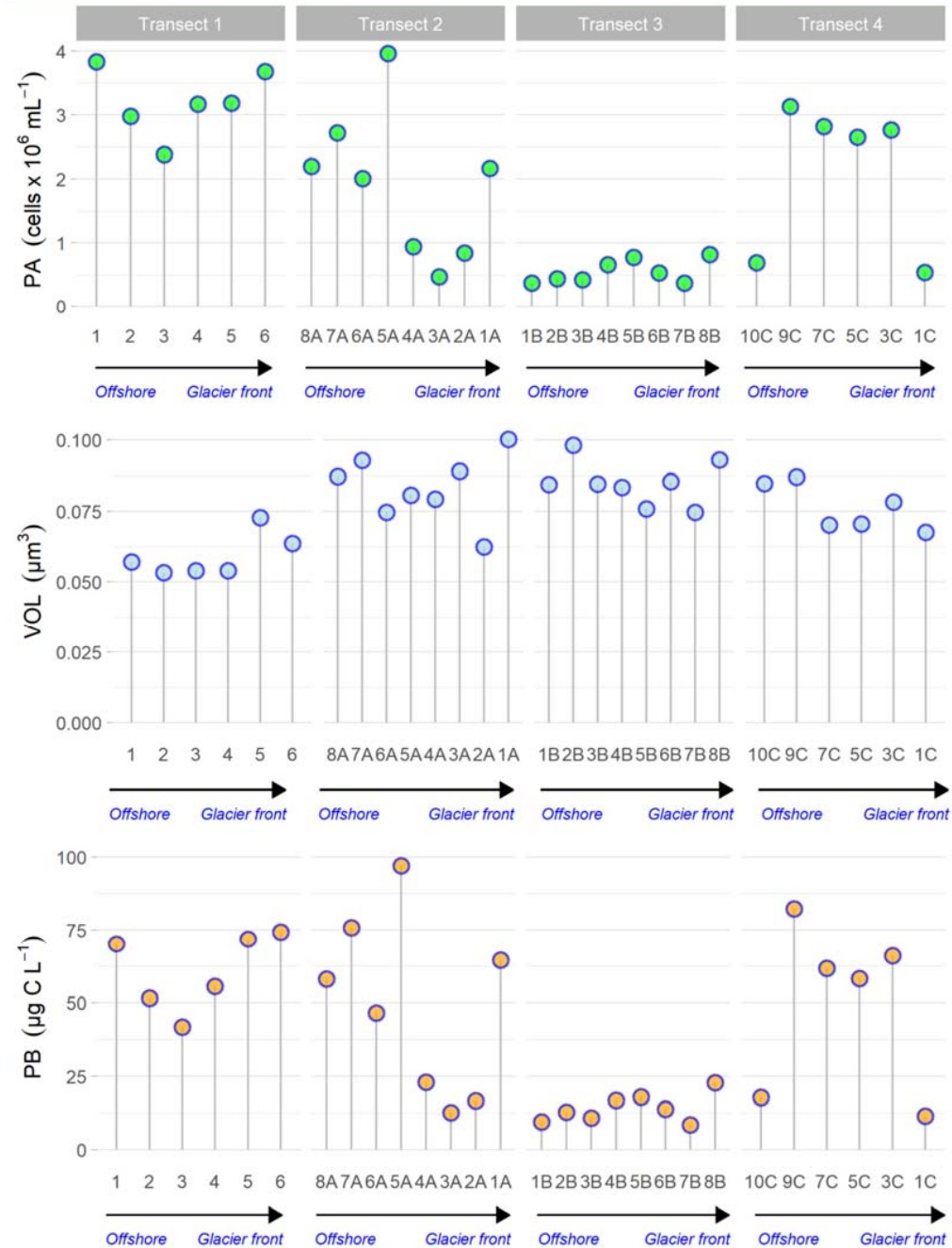

**Figure 4.** Spatial trends of prokaryotic abundance (PA), cell volumes (VOL) and prokaryotic biomass (PB) across the studied transects.

**Table 4.** Viable (Live) and respiring cells (CTC+) as percentages recorded across the four transects.

|  | Stations | Live | CTC |
|---|---|---|---|
|  |  | % | % |
| Transect 1 | 6 | 1.81 | 1.97 |
|  | 5 | 1.17 | 0.44 |
|  | 4 | 1.41 | 0.61 |
|  | 3 | 0.64 | 1.41 |
|  | 2 | 15.8 | 1.25 |
|  | 1 | 198 | 1.14 |

**Table 4.** *Cont.*

| | Stations | Live | CTC |
|---|---|---|---|
| | | % | % |
| | 1A | 0.99 | 3.09 |
| | 2A | 1.68 | 3.31 |
| | 3A | 2.07 | 4.48 |
| Transect 2 | 4A | 1.31 | 4.44 |
| | 5A | 0.51 | 4.2 |
| | 6A | 1.47 | 3.97 |
| | 7A | 3.21 | 2.89 |
| | 8A | 1.10 | 2.72 |
| | 8B | 0.70 | 2.90 |
| | 7B | 1.60 | 2.89 |
| | 6B | 1.09 | 2.97 |
| Transect 3 | 5B | 0.56 | 2.32 |
| | 4B | 0.91 | 2.52 |
| | 3B | 0.90 | 2.94 |
| | 2B | 5.84 | 2.28 |
| | 1B | 0.99 | 2.11 |
| | 1C | 20.8 | 2.17 |
| | 3C | 13.8 | 4.29 |
| Transect 4 | 5C | 0.96 | 1.48 |
| | 7C | 3.74 | 1.67 |
| | 9C | 0.89 | 2.42 |
| | 10C | 0.61 | 6.41 |

### 3.2.3. Virus-like Particles, Autotrophic and Prokaryotic Cells by Flow Cytometry

Spatial trends of virus-like particles across the transects were reported in Figure 5. The virus-like particle (VLP) abundance was in the order of $10^5$–$10^6$ VLP mL$^{-1}$. The highest counts were observed along Transect 4 (mean value $1.6 \pm 0.5 \times 10^6$ VLP mL$^{-1}$), while the smallest one was in Transect 3 ($0.53 \pm 0.14 \times 10^6$ VLP mL$^{-1}$). From the examination of the various stations in Transects 1, 2, 3 and 4, a clear variability was observed (CV = 47, 40, 27 and 32%, respectively). Four different viral sub-populations (namely V1, V2, V3 and V4) were distinguished based on their fluorescence signals, virus sub-population with the lowest green fluorescence being named V1 and the highest V4 (Figure 6). In all samples, the V1 group (mainly bacteriophages) accounted for 75% of the total VLP; a lower percentage was determined for the V2 group (17%), while V3 and V5 groups were negligible (5 and 2%, respectively). Virus to prokaryotic abundances ratios (VPR) ranged from 0.16 to 1.14 at different sampling points with similar average values among the Transects 1 (VPR = 0.38), 2 (VPR = 0.36) and 3 (VPR = 0.32). However, the average ratio VPR at Transect 4 (VPR = 0.88) showed significantly higher viral activity compared to the other transects.

Prokaryotic abundance by cytometry (PA$^C$) was in the order of $10^6$ cells mL$^{-1}$, with the highest abundances in Transect 1 (mean value $2.34 \pm 0.5 \times 10^6$ cells mL$^{-1}$) and the lowest in Transect 3 (mean value $1.73 \pm 0.2 \times 10^6$ cells mL$^{-1}$) in accordance with Image Analysis counts. Generally, the distribution in terms of abundance appeared to be homogeneous between the transects and stations (ANOVA). Prokaryotic cells with a high nucleic acid (HNA) content were always higher than LNA, accounting for an average value of 71% of the total cells (Figure 6). Picocyanobacteria were not observed by flow cytometry analysis. However, Picoeukaryote and Nanoeukaryote cells were present but in low abundance. Picoeukaryote abundances ranged from $9.23 \times 10^1$ to $9.95 \times 10^2$ cells mL$^{-1}$ with a homogeneous distribution among the transects, and Nanoeukaryote abundances ranged from $1.78 \times 10^1$ to $4.84 \times 10^2$ cells mL$^{-1}$ with greater abundances observed in Transects 2 and 4.

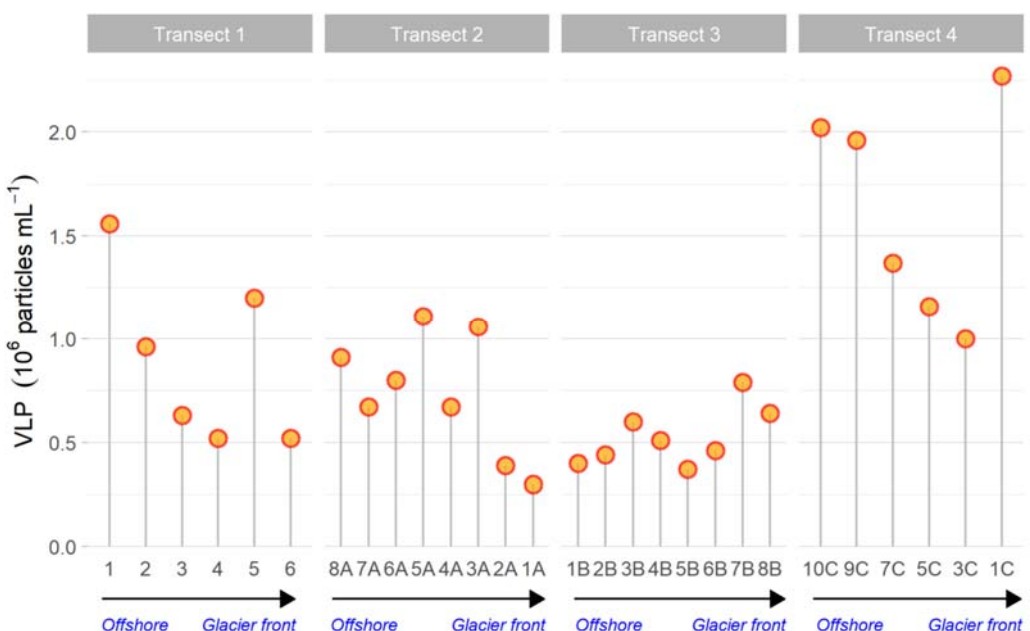

**Figure 5.** Spatial trends of virus-like particles across the transects.

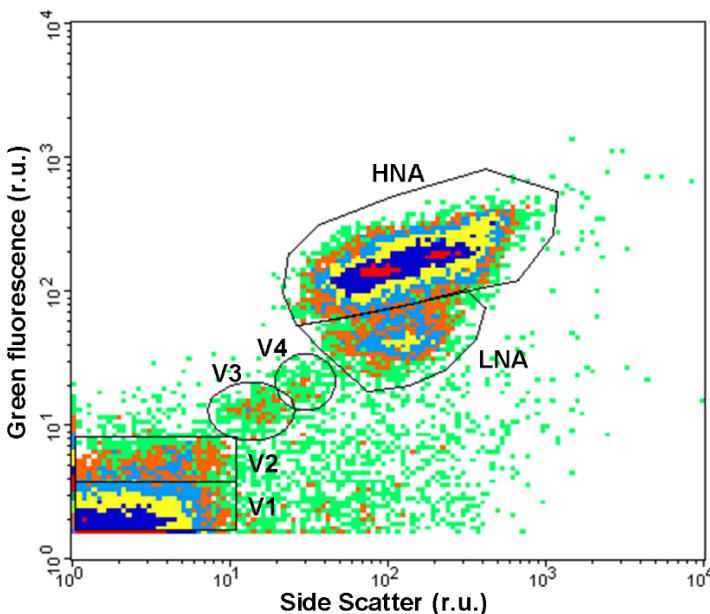

**Figure 6.** Flow cytometric analysis of surface sample after staining with Sybr Green I. The density plot shows in detail virus groups. V1, V2, V3, and V4, are distinguished according to their green fluorescence (V1 had the lowest fluorescence and V4 the highest), making up the total viruses. Prokaryotes with High DNA content (HNA) and Low DNA content (LNA) are also highlighted. All the data are expressed in r.u., relative units.

### 3.3. Culturable Heterotrophic Bacteria

The fraction of culturable heterotrophic bacteria was in the order of magnitude of $10^6$ CFU mL$^{-1}$. High abundances were found at Transects 1 and 2, with average values of $2.34 \pm 0.25 \times 10^6$ and $2.02 \pm 0.17 \times 10^6$ CFU mL$^{-1}$, respectively. Significant spatial variations were recorded by ANOVA (F = 11.00, $p = 9.78 \times 10^{-5}$).

### 3.4. Prokaryotic Community Composition

The relative abundance of bacterial groups was examined by CARD-FISH with bacteria-specific probes (EUB338 mix), an Archaea-specific probe (ARCH915) and 12 different bacterial group-specific probes (Table 1). The hybridisation level (percentage of DAPI-stained prokaryotes—i.e., Eubacteria plus Archaea—which were visualised by CARD-FISH) was 86.3%. With the set of probes used to detect the major divisions within the domain Bacteria, it was possible to affiliate 95.04% of the EUB338 hybridised cells to known bacterial groups. Overall, when considering probes targeting heterotrophic bacteria (Figure 7a), Proteobacteria (67.0% of EUB) were predominant and were followed by *Cytophaga–Flavobacterium* (CF) group of the Bacteroidetes (25.7%) and Cyanobacteria (2.0%). Overall, among Proteobacteria, Alpha- and Betaproteobacteria showed a comparable abundance in analysed samples (29.1 and 21.9%, respectively) for EUB-hybridized cells. Epsilon- and Deltaproteobacteria were less represented (with an average value of 0.35 and 0.34, respectively). Archaea, which were targeted by the probe ARCH915, represented, on average, 0.34% of the DAPI-stained cells.

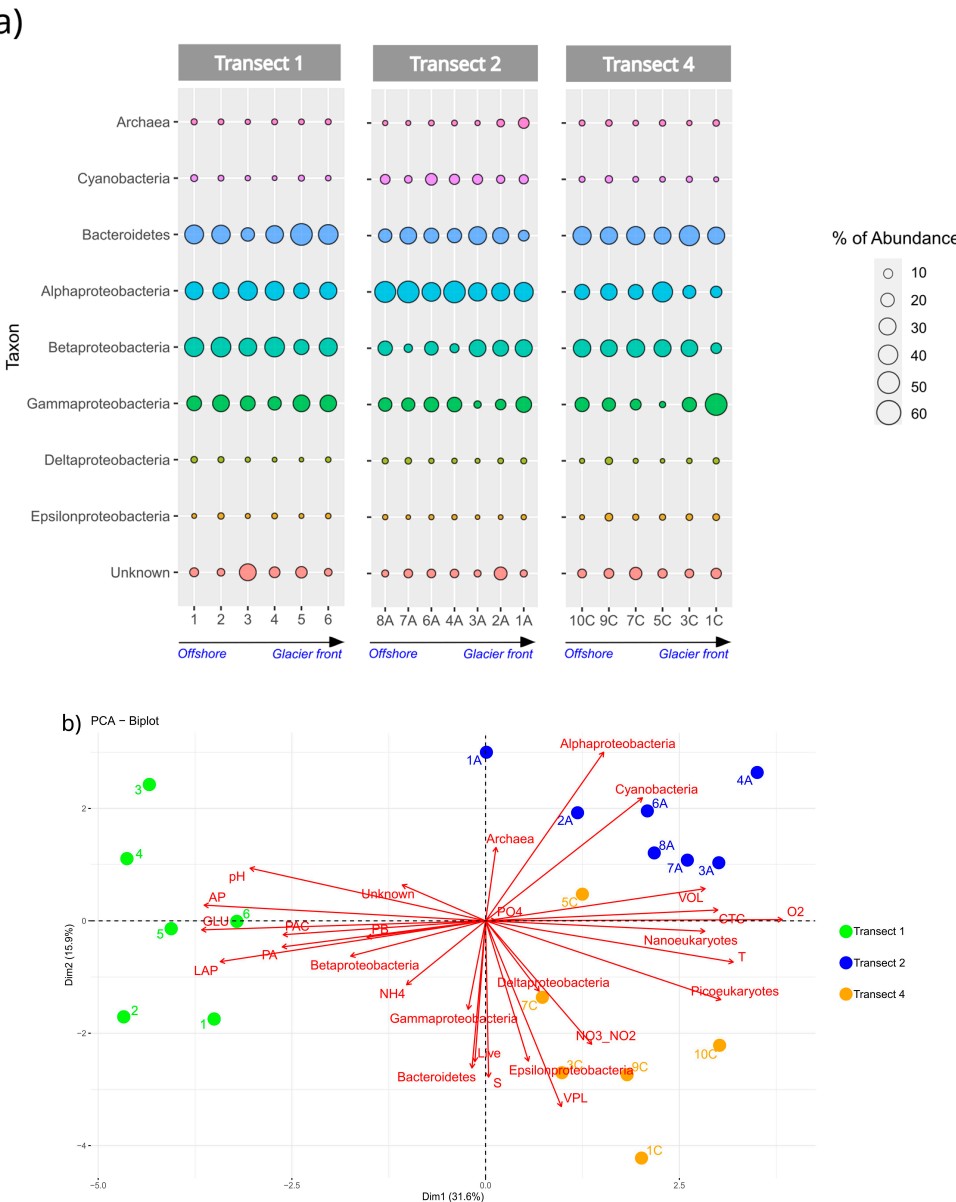

**Figure 7.** Results obtained from CARD-FISH analysis. (**a**) Bubble Plot showing the distribution of retrieved phyla in all analysed samples. The percentage was calculated on the total EUB probe count

for all bacterial phyla, whereas the archaeal percentage was obtained from the total DAPI count. (**b**) Principal Component Analysis (PCA), performed by R software using the package factoextra, computed on the physical, chemical and microbiological parameters for the three analysed transects.

A higher percentage of Proteobacteria (mainly Alpha-, Beta- and Gammaproteobacteria) was retrieved at offshore stations, with a decrease when approaching the glacier. Exceptions were stations 1A and 1C, which, despite being closest to the ice front, showed an overall proteobacterial abundance of 87.6 and 66%, respectively. Conversely, Bacteroidetes showed higher abundance values at the stations nearest to the glacier (e.g., stations 5, 3A, 2A and 3C with 40.0, 31.7, 23.8 and 42.0%, respectively). Also, in this case, stations 1A and 1C, compared to the rest of the transect, showed a different trend with an abundance of 7 and 27.1%, respectively. Cyanobacteria were found mainly along Transect 2, with their highest abundance at station 6A (9.1%). Finally, the abundance of Archaea was generally low and with comparable values (minimum value 0.1% at station 7A, maximum value 0.7% at station 9C) at all stations. Exceptions were stations 1A and 2A (the closest to the glacier in Transect 2), which showed abundances of 6.9 and 1.8% of Archaea, respectively.

The data from CARD-FISH, together with physicochemical data for the three analysed transects, were used to obtain a Principal Component Analysis (Figure 7b), highlighting how the transects were reciprocally separated into three almost totally distinct groups. It was also evident that along Transect 2, archaea, alphaproteobacteria and cyanobacteria prevailed, while Transect 4 was dominated by epsilon and deltaproteobacteria. Finally, Transect 1, which appeared to be the most different one, also showed Betaproteobacteria among its principal components.

The Spearman correlation analysis showed a positive correlation between Cyanobacteria and Alphaproteobacteria and $O_2$ ($p < 0.01$ and $p < 0.05$, respectively). Conversely, the Betaproteobacteria showed a negative correlation with $O_2$ ($p < 0.05$) and, at the same time, a positive correlation with the enzymatic activities AP and LAP, $p < 0.05$. The spatial variability of the phyla found was significant as determined by ANOVA when comparing the three transects (Transect 1 vs. Transect 2 and Transect 4, $p = 0.00865$ and $p = 9.46 \times 10^{-7}$, respectively; Transect 2 vs. Transect 4, $p = 0.0184$), showing however a clear distinction mainly between Transects 1 and 4.

*3.5. Microbial Metabolism*

3.5.1. Physiological Profiles of Microbial Community

The average substrate colour development (ASCD), expressed as absorbance values, is reported in Figure 8. High metabolic potentials were observed in Transect 3 ($0.201 \pm 0.09$), followed by Transect 1 (mean ASCD value: $0.188 \pm 0.101$) and Transect 2 ($0.187 \pm 0.09$) while, in Transect 4, the potential metabolism was low (mean ASCD values: $0.15 \pm 0.13$). The mean percentages of positive wells were high, accounting for 83, 73, 79 and 64% of the total wells per each plate in Transects 1, 2, 3 and 4, respectively.

Overall, the analysis of the functional diversity of the microbial community showed that the preferred carbon sources were those belonging to the groups of carbohydrates, carboxylic acids and Amino acids. On average, the microbial community had similar behaviour among the transects, except for Transect 2 (Supplementary Figure S1a), which, although carbohydrates were mostly used, differed from the others as it had higher degradation values towards phosphate-carbon and lower towards carboxylic acids. Analysing the results station by station, the utilised carbon sources were separated into two groups: in the first, carbohydrates, carboxylic acids and amino acids were widely utilised; in the second, complex carbon sources, phosphate-carbon and Amines were poorly utilised (Supplementary Figure S1b). In general, there was no trend correlating the geographic location of the samples with the preferred carbon source. However, considering the analysed stations, the separation into three large groups that preferred different carbon sources appeared evident. The first showed a high activity towards carbohydrates; a second group was characterised by the use of amino acids, carboxylic acids and, to a lesser extent, complex

carbon sources, in addition to carbohydrates; the last group included all the other samples showing intermediate characteristics between the two groups previously described.

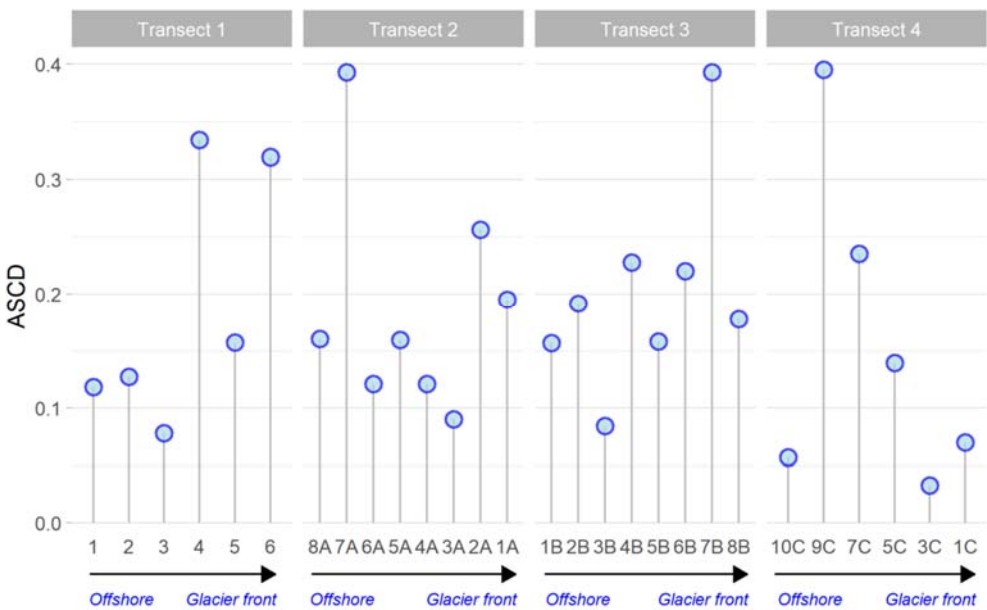

**Figure 8.** Average Substrate Color Development (ASDC) values determined of Biolog Ecoplates.

3.5.2. Enzymatic Activities

In all the transects, LAP prevailed compared to GLU and AP with average values of $1.928 \pm 1.954$, $0.496 \pm 0.602$ and $0.465 \pm 0.826$ nmol $L^{-1}$ $h^{-1}$ for LAP, GLU and AP, respectively (Figure 9). Spatial variability in the microbial metabolism was significant, as shown by ANOVA (F = 9.13, $p = 1.67 \times 10^{-3}$; 120.94, $1.56 \times 10^{-11}$; 80.92, $5.05 \times 10^{-10}$, for LAP, GLU, AP, respectively). At Transect 1, all the enzyme activities reached their peak values, with an average of $6.185 \pm 1.603$, $1.600 \pm 0.322$, and $2.249 \pm 0.586$ nmol $L^{-1}$ $h^{-1}$ for LAP, GLU and AP, respectively. For LAP and GLU, the highest activity rates were recorded at stations 1 and 2 (on the southern side of the transect), while for AP at stations 3 and 4 (in the middle of the transect), decreasing trends towards the glacier were observed. Along Transect 1, a minimum of LAP activity was recorded at station 3, in correspondence with a minimum of T. Transect 2 was characterised by the lowest enzyme activity levels, showing LAP, GLU and AP average values of $0.702 \pm 0.730$, $0.183 \pm 0.115$, and $0.049 \pm 0.084$ nmol $L^{-1}$ $h^{-1}$, respectively; similar average values were recorded at Transect 4. Spatially, LAP and AP were more active at coastal stations 1A and 2A and decreased along the transect; GLU followed an opposite pattern, with increased activity toward offshore stations. At Transect 3, microbial metabolism showed intermediate values between Transect 1 and Transects 2 and 4, with proteolytic and phosphatasic activity values about 3 and 2 times higher than those measured at Transect 2, respectively. High LAP and GLU activities were found at stations 2B and 3B, while AP peaked at station 3B in the middle of the transect. At transect 4, enzymatic values were similar to those measured at Transect 2.

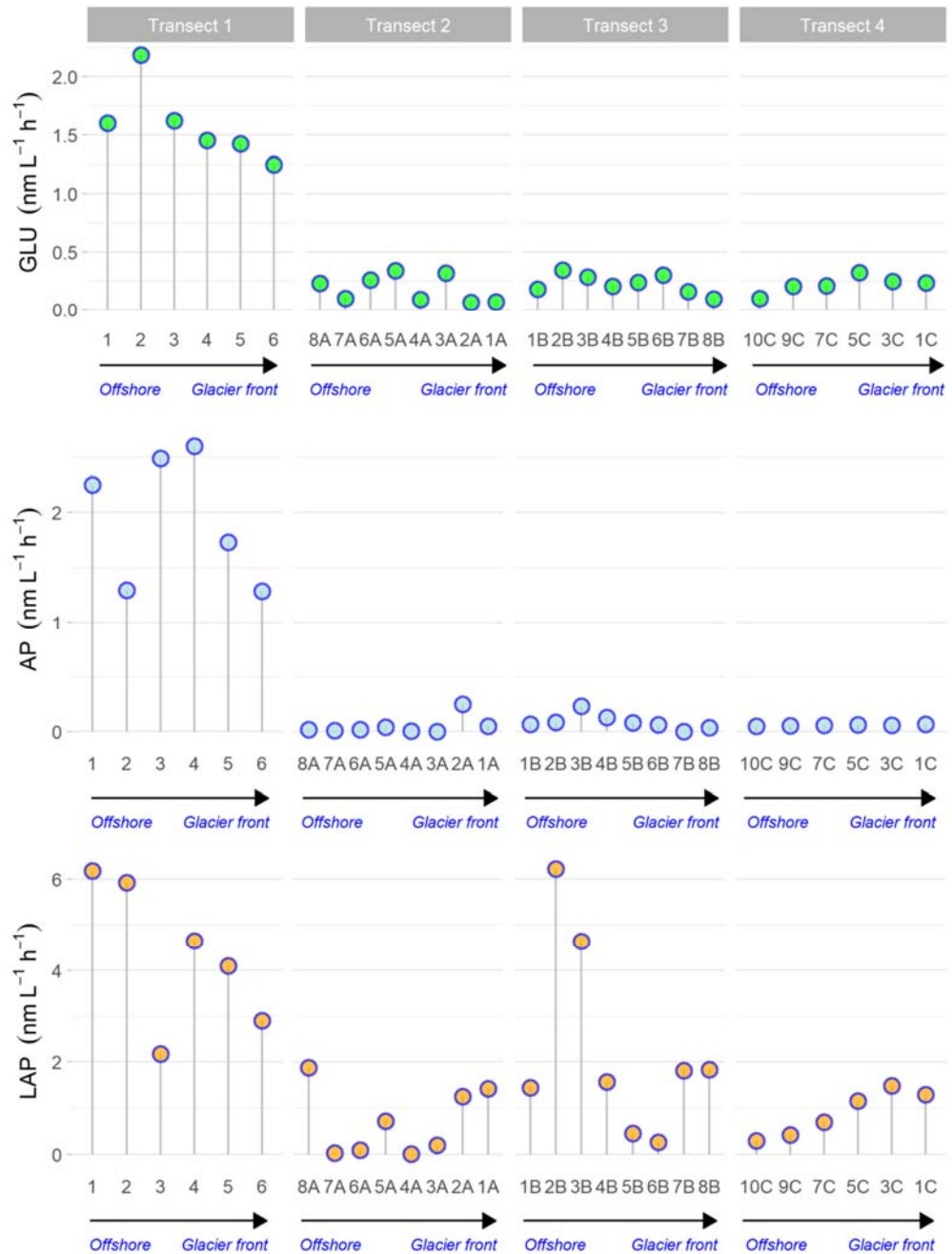

**Figure 9.** Spatial variations of enzyme activity rates [beta-glucosidase (GLU) of alkaline phosphatase (AP) and leucine aminopeptidase (LAP)] across the transects.

### 3.6. Statistical Analysis

Spearman correlation coefficients are shown in Figure 10. From the analysis, it was possible to highlight a strong negative correlation between the respiring cells, pico and nanoeukaryotes and enzymatic activities. Pico- and nanoeukaryotes are also strongly positively correlated among them. In relation to the physical and chemical parameters, respiring cells and Picoeukaryotes were negatively and positively related to the temperature, respectively, and there was a positive correlation between viral particles and salinity.

The results of the nMDS analysis (Figure 11a) highlighted how the sampling stations belonging to Transect 1 were totally separated from those of the other transects, and samples 1C and 10C clustered together in a separate group. Conversely, Transects 2, 3 and 4 showed a more homogeneous trend. From this figure, it is evident that the enzymatic activities (GLU, AP, and LAP) were the parameters responsible for the separation of Transect 1 from

the others. In the same way, the nanoeukaryotes' distribution seemed to guide the cluster formed by 1C and 10C samples.

Heatmap and cluster analysis (Figure 11b) also highlighted a distinction between the stations belonging to Transect 1 and those of the other transects, even if they clustered together with the stations along Transect 4; moreover, it highlighted how AP was the most distinctive parameter within Transect 1. Although it was not so evident, a grouping of the stations 7A, 5A, 1A, 8A and 6A, all belonging to Transect 2, was also observed. In the same figure, the separation of the group made up of the stations 1C and 10C was also evident. It should be noted that the parameters that stand out within our matrix were VLP, PA and PAC, which were completely separated from all the analysed parameters.

The PCA was made on the normalised data matrix to determine the weight of the parameters in the distribution of the sampling stations based on the variance calculation. The PCA (Figure 12) reconfirmed the above-reported results, highlighting the separation of Transect 2 from the others. Furthermore, stations 1A and 2A grouped together with Transect 2. In general, the total variance of the analysis was higher than 50% (53.3%), being distributed in Dim1 = 36.5% and Dim2 = 16.8%, respectively. Enzymatic activity, carbon source utilisation (ASCD) and pH were three distinctive parameters in quadrant I, whereas $O_2$ and VOL determined the sample distributions in quadrant II. The third quadrant distribution was instead guided by both T and biological parameters (CTC, Nanoeukaryotes, Picoeukaryotes and VPL). Finally, in quadrant VI, most of the influences came from nutrients ($PO_4^{3-}$ and $NH_4^+$), PA and S. The analysis first underlined the complete separation of transect 1 from the others, guided by pH and enzymatic activities. The stations belonging to Transects 2 and 4 were distributed over the II and the III quadrants, with a partial overlap; the 1C station was distinct from the remaining Transect 4 stations, which followed the behaviour of viral abundances. In the same way, station 1A also stands out by following the trend of the ASCD parameter. The PCA also highlighted the separation of the stations belonging to Transect 3, which were largely influenced by VOL and $O_2$ parameters.

Figure 13 shows the results of Maaslin2 analyses, or rather the statistically validated multivariable association. The Maaslin2 test results showed how the enzymatic activities represented the main difference in Transect 1 compared to the other transects (Figure 13a). Transect 3 is characterised by the parameters of CTC, VOL, PA and nanoeukaryotes (Figure 13b), showing significant differences with the other Transects, within which it is also evident how Transect 1 has lower values compared to the parameters mentioned above. Finally, Transect 4 (Figure 13c) showed a strong relationship with VLP, whereas Transect 3 is characterised by the lowest PA abundance.

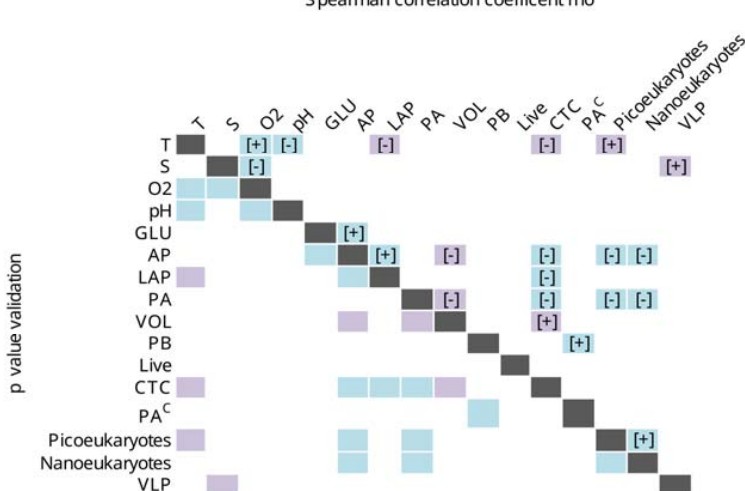

**Figure 10.** Spearman correlation calculated on all collected data along all analysed transects. In blue correlation with a *p*-value < 0.001, in violet correlation with a *p*-value < 0.005.

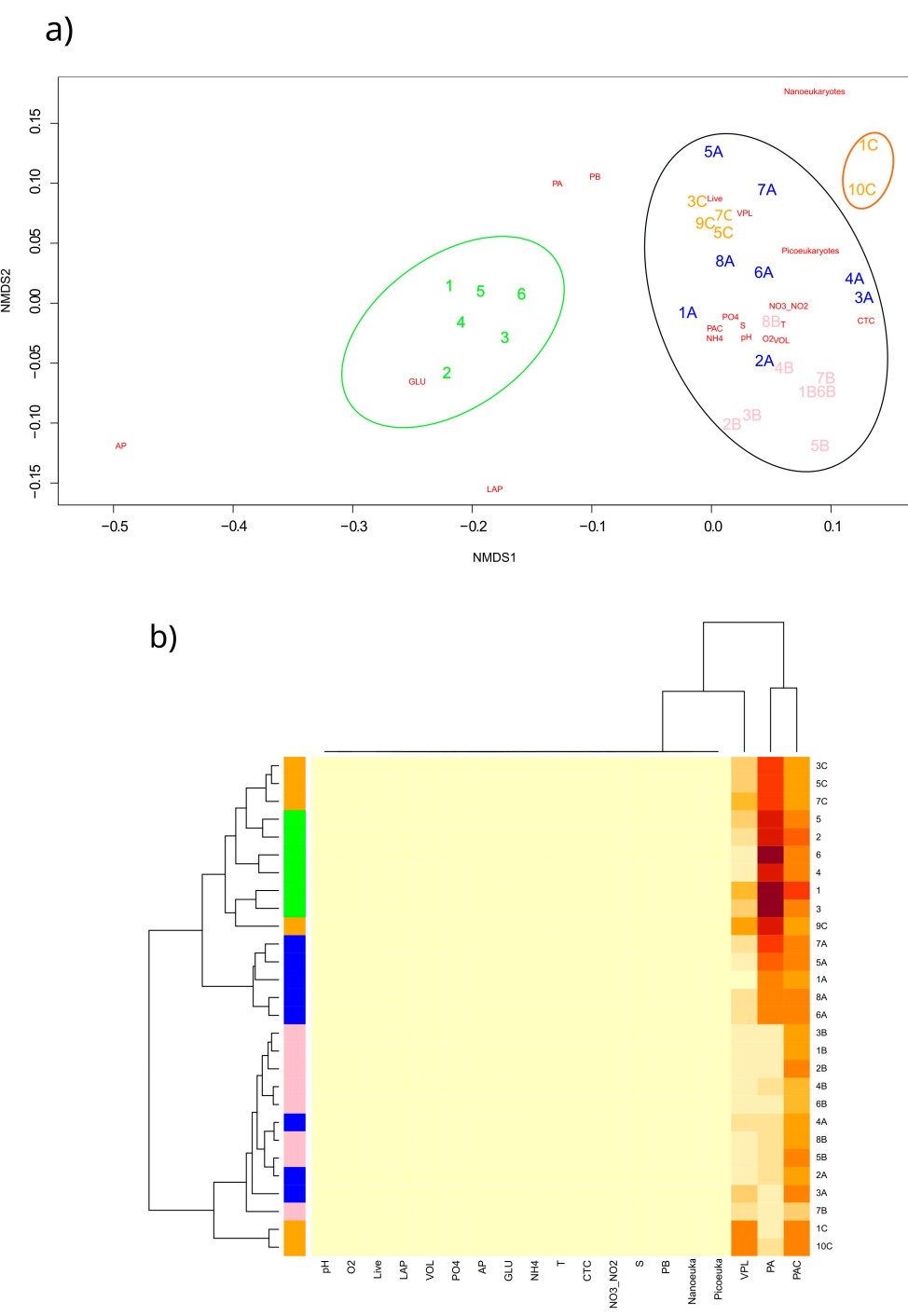

**Figure 11.** nMDS and Cluster analyses. (**a**) nMDS analysis, the stress was minimised after some number of reconfigurations of the points in 2 dimensions. The nMDS ordinations shown here were generated with k = 2 (linear fit r 2 = 0.921; non-metric fit r 2 = 0.982), and ellipses in the figure represent 80% of confidence between samples. (**b**) heatmap underlining how the investigated parameters varied across all analysed transects, their relation, and how the samples were grouped together.

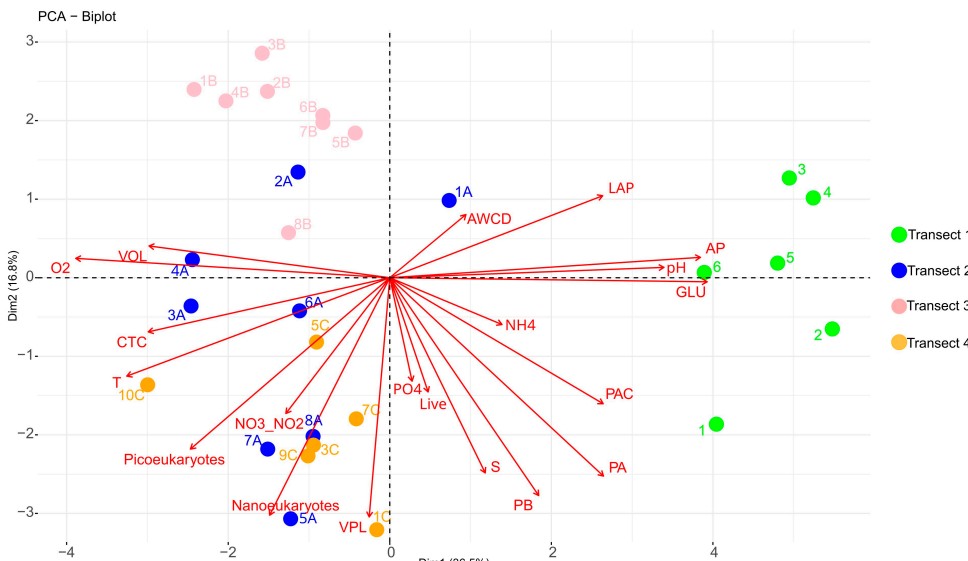

**Figure 12.** Principal Component Analysis (PCA), performed by R software using the package factoextra and computed on the entire dataset, including physical, chemical and microbiological parameters.

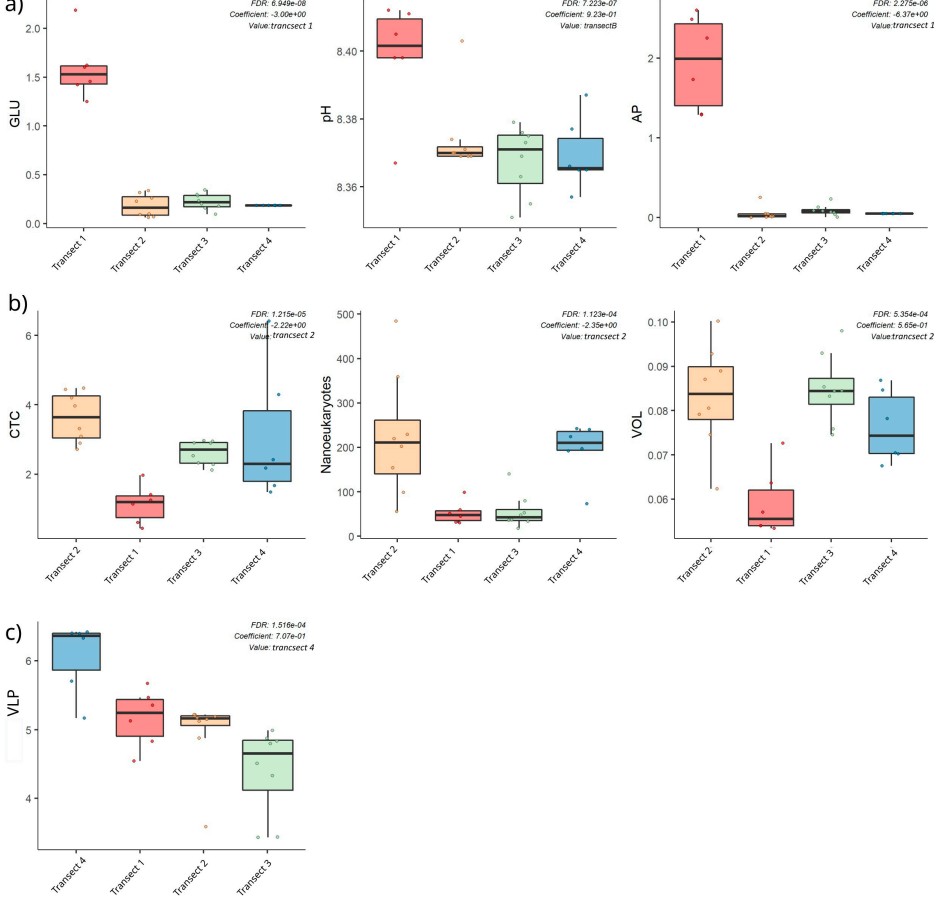

**Figure 13.** Significance results obtained from a comprehensive multi-model system for performing multivariable association testing in microbiome profiles (taxonomic, functional, or metabolomics) performed by Maaslin2 R packages. The figure is divided into three sections, with (**a**) showing the significant results obtained for Transect 1; (**b**) showing significant results obtained for Transect 2 and (**c**) showing significant results for Transect 4.

## 4. Discussion

The present survey was expressly designed to investigate the sea-glacier interface in terms of chemical, physical and biological parameters. The exploration of offshore-glacier transects, including the glacier melting area, increased the relevance of this study and is useful to deepen the knowledge of the dynamics of Kongsfjorden [6,11].

The development and field tests of automatic vehicles and water multi-samplers are of particular significance when planning research in extreme and hard-to-reach Arctic conditions. Furthermore, the modularity of PROTEUS makes it complementary to traditional oceanographic cruises, being a practical, low-cost system [43,44]. Besides, PROTEUS and the multi-sampler allowed the collection of water samples at precise points, avoiding external contamination with high qualitative standards for microbiological analyses.

Variations in temperature, salinity, oxygen, pH, nutrient cycling, and degradation of organic matter all influenced the microbial population [45]. In Kongsfjorden, the effects of melting glaciers on microbial communities due to the intrusion of freshwater are still not well understood, and few studies have been conducted on the characterisation of microbial communities in these environments [6].

Looking at the prokaryote counts (Figure 4), the abundances found in this study were comparable to data obtained by Jankowska et al. [46], who reported abundance/s in the range of $7.8–80.0 \times 10^5$ mL$^{-1}$ in two Arctic glacial fjords, but they were instead higher than those reported previously in Kongsfjorden [6,11,47–49]. The lowest abundances were obtained for Transect 3 with a mean value of $16 \times 10^5$ mL$^{-1}$.

To our knowledge, no data on the size of prokaryotic cells have been reported up to now for the Blomstrandbreen glacier. Here, the very small cell sizes (the mean total value of $0.077 \pm 0.01$ µm$^3$) prevailed (Figure 4) for all the analysed samples, but particularly in Transect 1, where cells reach a dimension smaller than $0.06 \pm 0.015$ µm$^3$. The occurrence of cell sizes smaller than 0.1 µm$^3$ has often been associated with life cycle changes, stressful situations and starved, dormant cells in extremely cold and salty environments [13,50,51], therefore in accordance with the dynamics of change of the investigated site. The biomass (Figure 4) widely varied, reaching values comparable to those reported for both temperate and polar marine environments [52,53]. Our estimations were higher than those reported in fresh waterbodies of Spitsbergen [54] and along a transect in Kongsfjorden [6]. In our study case, the abundance mainly contributed to biomass quantification than cell volumes.

Concerning prokaryotic viability, to our knowledge, no paper exists on Live/Dead cell counts and CTC in the Blomstrandbreen glacier. It is noteworthy that both technical procedures detected a low number of viable cells. In particular, estimates of living cells showed almost always a very low percentage (3.2% of the total L/D cells) with the exceptions of stations 3C and 1C (Transect 4) near the glacier, where the values rose to 14 and 21%, respectively. The average percentage of respiring cells with CTC ranged from 0.44 to 6.4% of the DAPI-stained bacteria. This observed percentage was similar to those reported in Kongsfjorden surface seawater samples and in Canadian Arctic surface freshwater samples [6,45]. Sherr et al. [55] found that a significant fraction of bacteria in the marine environment were dead or inactive. Similarly, in a study carried out in the marginal ice zone of the central Barents Sea [56], it was reported that a significant portion of the total bacteria was either dormant or expressed very low activity. Freshwater mixing following ice melt in the inner part of the Kongsfjorden has been found to act as an important factor in plankton mortality [57]. The Blomstrandbreen glacier is itself affected by active meltwater, with changes in the temperature, salinity and turbidity of the nearby water masses.

The VPR results (statistical proxy in the study of the virus-host relationship) [58–61] highlighted a high viral activity in Transect 4 (VPR 0.88) compared to the other three transects, which instead showed an average activity no higher than VPR $\cong$ 0.35 (Figure 6). The presence of several autotrophic microbial groups was also highlighted throughout the area. The absence of picocyanobacteria in the samples, as a result of microscopic analysis, could be explained by the fact that the presence of these microorganisms is uncommon in areas with temperatures below 8 °C [62]. On the other hand, our data

showed the occurrence of Picoeukaryote and Nanoeukaryote cells, even if they were in low abundance. Picoeukaryote abundances with a homogeneous distribution among transects, and Nanoeukaryote with greater abundances observed in Transects 2 and 4. In the literature, Pico- and Nanoeukaryotes are commonly described in ice-cold regions but with large variations in abundance and may not be observed or even reach a cell concentration of $1.82 \times 10^5$ mL$^{-1}$ [63]. The presence of Pico and Nanoeukaryotes, even at lower concentrations, indicates that water dilution by glaciers can affect cell abundance, but also demonstrates, together with viral and prokaryotic abundance data, that there is an intense microbial activity in the region [64,65].

If abundance and cell biomass provide numerous and important information, an essential datum is that on the metabolism and enzymatic activity of microbial populations. Microbial community metabolism by Biolog Ecoplates showed discrete potentials of carbon sources utilisation in the study area. The differences highlighted for ASCD (Supplementary Figure S1 and Figure 8) for the use of different substrates suggested the presence of diverse microbial communities (in terms of both abundance and composition) [45]. The response of each group of samples could be influenced by optimal growth temperature, incubation conditions and phylogenetic characterisation.

Considering the percentages of used substrates as an index of potential functional diversity, the best-exploited carbon sources in all the transects were carbohydrates. This is not surprising as they are the most abundant biomolecules on Earth, widely used by organisms for both structural and energy-storage purposes. Secondly, carboxylic acids were well utilised except for those in samples from Transect 2. These compounds, constituents of hydrophobic biomolecules, could derive from the catabolism of fatty acids and amino acids. This latter appeared to be oxidised uniformly throughout the study area. The adaptive capacity to change in relation to environmental constraints (glacier front, seawater influx, organic matter availability) caused modifications in the microbial community's physiological organisation, as has already been observed in other polar ecosystems [22,45,66].

Enzyme activity patterns recorded in the area of the Blomstrandbreen glacier were characterised by the predominance of LAP, suggesting the presence of labile substrates related to the productive summer period, while AP and GLU reached about 1/3 of the levels of LAP. In this study, the dominance of proteolytic activity compared to the other enzymes indicates that the microbial community was responsible for the quick mobilisation of N, as also shown by the inverse relationships observed between LAP, $NO_2^- + NO_3^-$ (as the enzyme reaction products). Microbial metabolism is generally modulated by both T, S and organic polymer availability [67]. In the entire study area, inverse relationships were observed between the enzyme activities and T (Figure 10), confirming the influence played by T on the expression of the measured enzymes [3,67].

Enzyme activity rates measured in this study were of the same order of magnitude as those reported during a previous study performed in an adjacent area of the Kongsfjorden across a central off-shore to coastal transect [6]. The distribution patterns of enzyme activities showed that spatially all the metabolic activities were particularly enhanced in Transect 1, suggesting the availability of organic substrates (i.e., phytoplankton exudates, zooplankton grazing, dead organisms, or terrestrial inputs), which provided a suitable trophic source and a substrate for enzymatic attack by the microbial community. This finding could also confirm the release into Arctic coastal waters of carbon- and nutrient-rich freshwater guided by ice-melt-driven terrestrial runoff, as reported by Delpech et al. [68]. Using all the obtained data, a principal component analysis was made to understand connection and correlation (Figure 12). The PCA highlighted the separation of Transects 1 and 2 from the others. Enzymatic activity and carbon source utilisation (ASCD) seem to be the distinctive characteristics of Transect 1, whereas, instead, Transect 2 seems to be distinct by $O_2$ and cellular volume. Interestingly, both sampling transects were further away from the glacier than the others. Transects 2 and 4 were instead partially overlapped, and their distribution was guided by chemicals, and physical parameters, particularly T and $NO_2^- + NO_3^-$ that also were significantly different between Transects 1 and 4 (df = 2;

F = 34.8; $p = 1.66 \times 10^{-9}$ and df = 2; F = 67.6; $p = 2.51 \times 10^{-7}$ respectively). All these data have been validated for significance using Maaslin2 analyses, and principal significant differences, also in this case, were found in Transects 1 and 3 (Figure 13).

Knowledge of marine bacterial populations can also be improved by establishing their taxonomy which can suggest their role and functions within a given habitat. Here we first report on the prokaryotic community composition at the Blomstrandbreen glacier-seawater front, made by CARD-FISH techniques. Results are in line with the biodiversity survey recently carried out in Kongsfjorden waters [69]. In fact, the authors report a simply structured community composition consisting of a few major bacterial taxa that were distributed in the fjord according to the water mass type, i.e., the hydrography of Kongsfjorden. In our study, even if with some exceptions, Proteobacteria and Bacteroidetes (as main taxa) followed opposite trends, with Proteobacteria abundances that generally decrease from the open sea to the glacier and Bacteroidetes that, instead, were more abundant at stations next to the glacier (Figure 7a). It is plausible to assume that Bacteroidetes could benefit from the probable presence of dissolved organic matter next to the glacier or derived from the glacier compartment [70]. Interestingly, the abundance trend of Proteobacteria was driven by different classes along the transects (Figure 7a), with mainly Alpha- and Betaproteobacteria. For instance, high abundances of Betaproteobacteria (56%) were found within prokaryotic communities in Arctic snow and ice [71,72]. Alphaproteobacteria are well-represented and are known as a prevalent taxon in the Arctic fjords, mainly in relation to algal blooms [73–76]. Archaea occurred at very low percentages at all stations, except for stations 1A and 2A (both along Transect 2), which were very close to the glacial front, suggesting a possible input from the glacial ice melting process to surface waters collected by PERSEUS. In fact, Archaea (with ammonium-oxidisers as the dominant group) have been previously detected on the surface of glaciers in snow and ice samples [77,78].

The data from CARD-FISH, together with physicochemical data, for the three analysed transects were used to obtain a principal component analysis (Figure 7b), highlighting how the transects clustered into three almost very distinct groups. It was also evident that Transects 2 and 3 (which were the nearest to the glacier) had a greater similarity, probably depending on their location. In fact, these two transects partially overlapped and were the closest to the glacier (station 1A was 6.7 m from the glacier, and 1C was 0.8 m from the glacier). Transect 1, which appears to be the most different one, seems to be the most active, also based on enzymatic activities, and was the farthest from the glacier front (station 6 was 286.2 m from the glacier). This finding could be dependent on glacial melting, which can start internal water recycling and circulation, generating and modifying the physical-chemical parameters essential for microbial life [7,46]. More in-depth studies will be devoted to deepening the information gained in this study on prokaryotic community composition at the glacier-seawater front.

## 5. Conclusions

The use of remotely controlled technologies has been proven to be a successful tool in enabling the collection of seawater samples for microbiological analyses. The obtained data furnish an updated overview of the abundance, structure and metabolism of microbial communities in a difficult-to-access area of the Svalbard archipelago and how they respond to environmental variables.

Multivariate analysis of microbial communities highlights some key trends/features:

- Transect 1 was totally different from the others (2, 3, 4) that were mostly well-characterised in their distinctive properties (even if with some anomalous samples). The regression also confirmed these significant differences between Transect 1 and the others, as well as between Transects 3 and 4;
- Transect 2 was characterised by large-sized microorganisms, actively respiring cells and terrigenous inputs (i.e., Nanoeukaryotes, VOL, CTC, $PO_4^{3-}$);

— In Transect 4, the microbial community appeared to be controlled by viruses and characterised by small-sized cells (Picoeukaryotes) due to the reduced temperature and dilution of glacier water (low salinity);

— Differences in microbial abundance and metabolism were attributable to complex environmental conditions like ice melting or Atlantic water inflow.

**Supplementary Materials:** The following supporting information can be downloaded at: https://www.mdpi.com/article/10.3390/w15030556/s1, Supplementary Figure S1. Biolog Ecoplate results expressed as percentage of used carbon source groups. Carbon sources were grouped as follow: Complex carbon sources (Tween 40, Tween 80, α-cyclodextrin, Glycogen); Carbohydrates (D-cellobiose, α-D-lactose, β-methyl-D-glucoside, D-xylose, i-erythritol, N-acetyl-D-glucosamine, D-mannitol); Phosphate-carbon (Glucose-l-phosphate; D,l-α-glycerol phosphate); Carboxylic acids (Pyruvic acid methyl ester, D-glucosamic acid, D-galactonic acid γ-lactone, D-galactonic acid, 2-hydroxy benzoic acid, γ-hydroxy butyric acid, 4-hydroxy benzoic acid, Itaconic acid, α-ketobutyric acid, D-malic Acid); Amino acids (L-arginine; L-asparagine; l-phenylalanine; L-serine; L-threonine; Glycyl-L-glutamic acid); Amines (Phenylethyl-amine; Putrescine). percentage of carbon sources used, calculated on each sampling point.

**Author Contributions:** Conceptualisation, M.A. and G.B.; methodology, M.P., G.M., A.C.R., M.A., F.A., G.C., A.L.G., R.P., A.S.C., R.LF., R.F., A.C., A.O., G.Z., G.B. and M.C.; software, M.P. and A.C.; validation, M.P., G.M., A.C.R., M.A., F.A., G.C., A.L.G., R.P., A.S.C., R.L.F., R.F., A.C., A.O., G.Z., G.B. and M.C.; formal analysis, R.L.F. and A.L.G.; investigation, M.A. and G.B.; resources, M.A. and G.B.; data curation, M.P., G.M., A.C.R. and A.C.; writing—original draft preparation, all the authors; writing—review and editing, all the authors; visualisation, all the authors; supervision, M.A. and G.B.; project administration, M.A. and G.B.; funding acquisition, M.A. and G.B. All authors have read and agreed to the published version of the manuscript.

**Funding:** This research was funded by the "Arca" (M.A.), UVASS (G.B.), CASSANDRA (PRA; M.A.) and RESTORE projects (PNRA; G.B., M.A.). R.L.F. was supported by the National Council of Research in the frame of the Short Term Mobility program (STM- AMMCNT. CNR prot. n. 0070784/2019 del 15/10/2019); RP (Rodolfo Paranhos) CNPQ 312.479/2020-4 & Faperj 201.112/2021; ASC by the PNPD/CAPES 88887.466809/2019-00.

**Data Availability Statement:** Data generated or analysed during this study are contained within this article.

**Acknowledgments:** Tanks are due to Anderson Aquino, Laboratory of Hidrobiologia of UFRJ (Brazil), for his technical assistance in the flow cytometry analyses.

**Conflicts of Interest:** The authors declare no conflict of interest. The funders had no role in the design of the study; in the collection, analyses, or interpretation of data; in the writing of the manuscript, or in the decision to publish the results.

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
