# Peer review of "Microbial Community Abundance and Metabolism Close to the Ice-Water Interface of the Blomstrandbreen Glacier (Kongsfjorden, Svalbard): A Sampling Survey Using an Unmanned Autonomous Vehicle"

_water, doi:10.3390/w15030556_

Round 1
Reviewer 1 Report (New Reviewer)
In light of the quantity of samples collected and analyses performed, I believe this article is worthy of publication; however, more effort may be made to organize the discussion paragraph, which, in my opinion, frequently lacks incisiveness. The sampling campaign could be properly explained by the authors, and it should be made clear which distinctions exist between the transects. Some results are not statistically significant at all, so they can be included as supplementary material. Authors conclude that differences were recorded among the transects probably due to ice melting or Atlantic water inflow, but in my opinion differences between the transects, which are supported by statistical analyses (Figs 11, 12, and 13), should be discussed in greater detail.

Author Response
Referee 1
Review Report Form
Open Review
Comments and Suggestions for Authors
In light of the quantity of samples collected and analyses performed, I believe this article is worthy of publication; however, more effort may be made to organize the discussion paragraph, which, in my opinion, frequently lacks incisiveness. The sampling campaign could be properly explained by the authors, and it should be made clear which distinctions exist between the transects. Some results are not statistically significant at all, so they can be included as supplementary material. Authors conclude that differences were recorded among the transects probably due to ice melting or Atlantic water inflow, but in my opinion differences between the transects, which are supported by statistical analyses (Figs 11, 12, and 13), should be discussed in greater detail
Thank you very much for the very constructive comment. Following your comments, the authors have revised and restructured the discussions with attention to the statistically significant results, added more information on the sampling plan and inserted in supplementary material, the superfluous. The conclusion now reported in detail the microbial differences between the transects.

Reviewer 2 Report (New Reviewer)
The manuscript was studied to investigate the sea-glacier interface in terms of chemical, physical and biological parameters. A novel and promising technical approach was the adoption of the remotely controlled PROTEUS sampler, a new autonomous vehicle prototype. In addition, The samples were analysed for: abundance of total prokaryotes, viable and respiring cells, their morphological traits and biomass by Image Analysis; autotrophic and prokaryotic cells (with high and low nucleic acid contents) as well as virus-like particle counts by flow cytometry; potential community metabolism by BIOLOG ECOPLATES; and potential enzymatic activity rates on organic polymers by fluorimetry. The analysis is thorough, and the result part is sufficient. I think the topic is of interest and better understandable. However, some issues still need to be explained and revised by the authors.
Major comments
Point 1: the information would contribute to predicting the microbial community abundance and metabolism close to the ice-water interface of the Blomstrandbreen glacier. However, I think this manuscript has not been established as a research article well and includes many redundant words. The authors have to redraft entire parts of this manuscript to improve this to be more understandable. The abstract is not established well, and the authors should revise it to focus on the main findings. In this manuscript, some figures need to move in supplementary data. The authors did not mention the figure or table names in the different paragraphs in the results section, which was quite confusing.
Point 2: Difficult to understand the logical relationship in the introduction section. Please improve. Another weakness of this manuscript is the clarity of the writing. Many sentences are redundant, grammatically incorrect or simply awkward or unclear. It is difficult for the reader to evaluate the validity of the authors’ conclusions because of the lack of clarity in the writing.
Minor comments
Line 60: “whole ecosystem” not clear, please clarify.
Line 64-65: mention specific citations of each (ice-free and ice-covered surface waters; temperature approaching freezing point values at depth layers, extreme seasonality, and strong light variations).
Line 69: hampering? What is this
Line 118-120: an odd sentence. Please clarify.
Line 126: Figure 1, the picture quality is very low. If possible, please provide high-resolution pictures.
Line 132-135: If possible, please write in descending orders (1C…10C), (1A…8A).
Line 142-144: Difficult to understand. Please revise this sentence.
Line 150 mentions three replicates, and line 157 two replicates. Confused? Please clarify.
Line 163: “as previously described [22]” mention the author name or journal format not sure.
Line 168-170: an odd sentence. Please revised.
Line 197-200: read many times, still not clear what leads to this sentence. Please improve the logic of this sentence.
Line 231: needs to be revised.
Line 234: Please cross-check the equation.
Line 380-382: I think it should be moved to the result section.
Line 432: “p<0.05” I recommend changing to “P < 0.05” throughout the manuscript.
Line 467: Figure 9a, I think it moved to supplementary data.
Line 501-508: moved to material and methods.
Line 509-510: Table 3? please specify where is the table? otherwise, this statement is not meaningful.
Line 511-524: These sentences' structure is relatively weak and does not represent the results. Please revised.
Line 549-552: odd, please revise.
Line 671: odd, revised as a scientific language. Please improve the logic overall in the discussion section.
Line 699-707: revised the conclusion.
Please ask all authors (experts) to read, revise and polish the redundant sentences of the manuscript before submitting.
Author Response
Referee 2
Comments and Suggestions for Authors
The manuscript was studied to investigate the sea-glacier interface in terms of chemical, physical and biological parameters. A novel and promising technical approach was the adoption of the remotely controlled PROTEUS sampler, a new autonomous vehicle prototype. In addition, The samples were analysed for: abundance of total prokaryotes, viable and respiring cells, their morphological traits and biomass by Image Analysis; autotrophic and prokaryotic cells (with high and low nucleic acid contents) as well as virus-like particle counts by flow cytometry; potential community metabolism by BIOLOG ECOPLATES; and potential enzymatic activity rates on organic polymers by fluorimetry. The analysis is thorough, and the result part is sufficient. I think the topic is of interest and better understandable. However, some issues still need to be explained and revised by the authors.
Major comments
Point 1: the information would contribute to predicting the microbial community abundance and metabolism close to the ice-water interface of the Blomstrandbreen glacier. However, I think this manuscript has not been established as a research article well and includes many redundant words. The authors have to redraft entire parts of this manuscript to improve this to be more understandable. The abstract is not established well, and the authors should revise it to focus on the main findings. In this manuscript, some figures need to move in supplementary data. The authors did not mention the figure or table names in the different paragraphs in the results section, which was quite confusing.
In this revised version, the authors have deleted the redundant words as well as sentences to make the manuscript clear and more fluent. The abstract has been improved by redrafting. The main findings of the study have been described in details in “Conclusions” section.
Point 2: Difficult to understand the logical relationship in the introduction section. Please improve. Another weakness of this manuscript is the clarity of the writing. Many sentences are redundant, grammatically incorrect or simply awkward or unclear. It is difficult for the reader to evaluate the validity of the authors’ conclusions because of the lack of clarity in the writing.
The “Introduction” section has been modified to ameliorate the logical development and to correct the grammatical mistakes. Conclusion have been punctually described.
Minor comments
Line 60: “whole ecosystem” not clear, please clarify.
The entire introduction have been rewritten according to all the referee comments
Line 64-65: mention specific citations of each (ice-free and ice-covered surface waters; temperature approaching freezing point values at depth layers, extreme seasonality, and strong light variations).
The specific citations are mentioned at the end of the sentence “[4,5]”
Specifically:
- Alonso-Sáez, L; Sánchez, O.; Gasol, J.M..; Balagué, V.; Pedrós-Alio, C. Winter-to-summer changes in the composition and single-cell activity of near-surface Arctic prokaryotes. Microbiol. 2008, 10(9), 2444-2454. https://doi.org/10.1111/j.1462-2920.2008.01674
- Müller, O.; Wilson, B.; Paulsen, M.L.; Ruminska, A.; Armo, H. R.; Bratbak, G.; Øvreås, L. Spatio temporal dynamics of ammonia-oxidizing Thaumarchaeota in distinct Arctic water masses. Microbiol. 2018, 9, 24. https://doi.org/10.3389/fmicb.2018. 00024
Line 69: hampering? What is this
We substituted this word
Line 118-120: an odd sentence. Please clarify.
We rearranged the sentence
Line 126: Figure 1, the picture quality is very low. If possible, please provide high-resolution pictures.
We repaint the figure.
Line 132-135: If possible, please write in descending orders (1C…10C), (1A…8A).
Thanks, but the order we used is guided by the collecting strategy during the multidisciplinary project. The authors think is higher useful to leave this order.
Line 142-144: Difficult to understand. Please revise this sentence.
This sentence has been revised
Line 150 mentions three replicates, and line 157 two replicates. Confused? Please clarify.
Thanks, we corrected it
Line 163: “as previously described [22]” mention the author name or journal format not sure.
The entire manuscript has been checked and fixed up for these typos
Line 168-170: an odd sentence. Please revised.
The sentence has been corrected
Line 197-200: read many times, still not clear what leads to this sentence. Please improve the logic of this sentence.
We rearranged the sentence
Line 231: needs to be revised.
Done (lines 191-193)
Line 234: Please cross-check the equation.
According to Sala et al. (2005), the color development for each plate was expressed as the average substrate color development (ASCD). The formula adopted was by Sala et al. (2006) i.e. ASCD was calculated after subtracting the average absorbance of the control wells (C, without a carbon source) from the average absorbance of each well (R, with a carbon source) and then calculating the mean absorbance of each substrate of the 3 replicate wells for each substrate (31).
Line 380-382: I think it should be moved to the result section.
We include these results also in the text (lines 365-368), anyway we left the sentence in the figure legend because is useful for the figure’s understanding.
Line 432: “p<0.05” I recommend changing to “P < 0.05” throughout the manuscript.
The entire manuscript has been changed for these typos
Line 467: Figure 9a, I think it moved to supplementary data.
The figure has been moved to the supplementary data.
Line 501-508: moved to material and methods.
Thanks but, these are the results of the spearman correlation, which is already mentioned in the materials and methods paragraph.
Line 509-510: Table 3? please specify where is the table? otherwise, this statement is not meaningful.
The table has been changed in figure 10
Line 511-524: These sentences' structure is relatively weak and does not represent the results. Please revised.
Thanks, we revised the sentence
Line 549-552: odd, please revise.
We rearranged the sentence
Line 671: odd, revised as a scientific language. Please improve the logic overall in the discussion section.
The section “discussion” has been improved
Line 699-707: revised the conclusion.
The conclusion has been rearranged
Please ask all authors (experts) to read, revise and polish the redundant sentences of the manuscript before submitting.
All authors have been revised the manuscript and polish the redundant sentences
Round 2
Reviewer 1 Report (New Reviewer)
Dear authors, I have read the final version and I have not additional recommendations. I have given my approval to its publication.
This manuscript is a resubmission of an earlier submission. The following is a list of the peer review reports and author responses from that submission.
Round 1
Reviewer 1 Report
I find the article confusing. The authors stress the fact that their bespoke sampling robot is the novelty of the article. As a reader, it feels as if the microbiology results are secondary. Certainly the results do not support the stated aim of understanding the impact of climate change (if that is indeed the aim, which is not at all clear, even though this is the aim stated in the abstract: "...survey...to get novel insights into this biological component [sic] and its response to climate change").
What the paper has reported is the microbiological variability of four (or three), geographically close, sampling transects below the Bloomstrandbreen glacier. It is important not to overstate the significance or consequence of the work. The first sentence of the Discussion section indicates the authors are aware of the real value.
On another note, the idea that Ny-Alesund is remote is odd, given that across the bay, there is a relatively luxurious international station.
Finally, much of the grammar of the paper makes it difficult to read, and in places difficult to understand what is meant.
Reviewer 2 Report
The paper by Maria Papale and colleagues describes microbial community features of the Kongsfjorden at the Western Spitsbergen coast. Overall the paper is well composed and it has a clear conclusion, however there are also some shortcomings that need to be addressed. First of all the quality of the English language is somewhat lacking. It seems like parts of the paper are written in good English and some are worse. This needs to be corrected by a native speaker preferably.
Specific comments (verses):
22 - severely affecting severely (?)
23 - effects played or exerted?
25 – Blomstrandbreen glacier - breen means glacier, so either Blomstrandbreen or Blomstrand glacier here and throughout the manuscript
26 - There is now such entity as the Arctic Sea, it's either Arctic Ocean or the in the case of Western Spitsbergen - the Greenland Sea
43 - how is this forbidden? Please find a better term.
57-59 - very confusing. Is this permafrost mention necessary?
63 - inter-moraine depressions
114 – 2 8 a very confusing notation, change to 2 – 8 or 2 – eight, here and further in the text
123 - here and throughout the manuscript compounds like nitrates etc. should be presented as ions
169 - photoautotrophic rather than autotrophic if chlorophyll presence was the defining feature
207 - the carbon sources are not oxidized into formazan, the formazan forms as a result of bacterial respiration with carbon compounds serving as energy source
306 - explain CV upon the first mentioning
Figure 7A - those are not only phyla, there are also classes; change phyla to taxon
649-650 - inflow by what? a paper?
696-719 this summary has no standing here, place it in the Results section or the Conclusions